# Progress and Recent Strategies in the Synthesis and Catalytic Applications of Perovskites Based on Lanthanum and Aluminum

**DOI:** 10.3390/ma15093288

**Published:** 2022-05-04

**Authors:** Helir Joseph Muñoz, Sophia A. Korili, Antonio Gil

**Affiliations:** Departamento de Ciencias, Universidad Pública de Navarra, Campus de Arrosadía, 31006 Pamplona, Spain; helirjoseph.munoz@unavarra.es (H.J.M.); sofia.korili@unavarra.es (S.A.K.)

**Keywords:** perovskite, lanthanum aluminate, synthesis methods, catalytic applications

## Abstract

Lanthanum aluminate-based perovskite (LaAlO_3_) has excellent stability at high temperatures, low toxicity, and high chemical resistance and also offers wide versatility to the substitution of La^3+^ and Al^3+^, thus, allowing it to be applied as a catalyst, nano-adsorbent, sensor, and microwave dielectric resonator, amongst other equally important uses. As such, LaAlO_3_ perovskites have gained importance in recent years. This review considers the extensive literature of the past 10 years on the synthesis and catalytic applications of perovskites based on lanthanum and aluminum (LaAlO_3_). The aim is, first, to provide an overview of the structure, properties, and classification of perovskites. Secondly, the most recent advances in synthetic methods, such as solid-state methods, solution-mediated methods (co-precipitation, sol–gel, and Pechini synthesis), thermal treatments (combustion, microwave, and freeze drying), and hydrothermal and solvothermal methods, are also discussed. The most recent energetic catalytic applications (the dry and steam reforming of methane; steam reforming of toluene, glycerol, and ethanol; and oxidative coupling of methane, amongst others) using these functional materials are also addressed. Finally, the synthetic challenges, advantages, and limitations associated with the preparation methods and catalytic applications are discussed.

## 1. Introduction

The term perovskite is currently used to refer to the extensive family of materials whose crystal structure is based on that of the mineral CaTiO_3_ [1,2,3,4]. Perovskites show the general formula ABO_3_, where A and B are cations with different sizes [5]. The atomic size of the atoms in position A is larger than the atoms in position B [6], and these can present isomorphic substitutes by semimetals or metal. In all cases, the anionic position of the perovskite is occupied by oxygen atoms [5]. Interestingly, perovskites can have different compositions due to the partial substitutions of cations A or B by other cations with the same or different oxidation states, thus, the general chemical formula of perovskite compounds is as follows: A_1−x_A′_x_B_1−y_B′_y_O_3±δ_ [7].

The wide range of perovskites available include lanthanum aluminate perovskite (LaAlO_3_), which presents a phase transition from a rhombohedral crystal structure (space group R-3c) to cubic (space group Pm-3m) at approximately 525 °C [8]. This material has an excellent thermal stability, high transparency, and an optical band gap similar to that for semiconductors (from 5.6 to 7 eV) a high dielectric constant and also presents high chemical resistance and low toxicity [8,9]. LaAlO_3_ perovskite is also an interesting material because of its excellent ability to promote a synergistic effect on different metal catalysts, such as Ni, Co, Pt, and Zn, amongst others [10]. 

Recently, due to its important characteristics, LaAlO_3_ perovskite has gained importance due to its possible application as a catalytic, electronic, optic, and luminescent material [11]. For example, LaAlO_3_ has been used as a catalyst solid for the hydrogenation and hydrogenolysis of hydrocarbons [12,13], as a nano-adsorbent [14], for the oxidative degradation of organic pollutants [15], and as a substrate for different thin films [16,17], amongst other applications.

Recent studies have shown that the crystalline structure of LaAlO_3_ is a good host for +3 cations as long as these have an ionic radii similar to that of the La^3+^ ion, different rare earth ions meet this requirement, thus, allowing this perovskite to incorporate high concentrations of these ions (up to 20%) into their crystalline structure [8]. Furthermore, the B-site occupied by aluminum can be doped with transition metal ions, thus, resulting in an extensive LaAlO_3_ family. LaAlO_3_ perovskite has been doped with rare-earth ions, such as Tb^3+^, Eu^3+^, and Tm^3+^; as well as noble metal ions, such as Rh^3+^, Ru^3+^, Ir^3+^, Pd^2+^, and Pt^4+^; or with Ca^2+^, Ni^2+^ Co^2+^, Mn^2+^, Fe^3+^, and Ga^2+^, amongst others. 

The resulting materials have been investigated in a wide variety of energetic applications, for example as catalysts in the dry and steam reforming of methane (DRM and SRM, respectively) as well as in the steam reforming of toluene, glycerol, and ethanol and the oxidative coupling of methane (OCM) [9,18,19]. They have also been studied as three-way catalysts (TWC) [20], as electrocatalysts [21], and as luminescent material [8]. In other research, the optical properties of rare-earth-doped LaAlO_3_ matrices have been analyzed by thermoluminescence using ultraviolet radiation and X-ray dosimetry [8]. 

In this sense, recent extensive studies have focused on optimizing the synthetic methods for LaAlO_3_ perovskites to prepare finer, more homogenous nanopowders, to improve their catalytic and optoelectronic characteristics or to induce new characteristics in these materials [8,13]. This optimization has resulted in improvements in the protocols and parameters of the various preparation methods, such as shorter heat-treatment periods, control of reaction by-products, and calcination at lower temperatures, amongst other factors [8,9,10,12,13,18].

This review work covers the main strategies developed over the past 10 years to synthesize some representative members of the LaAlO_3_ family in order to provide the reader with a rational guide to understand the synthetic challenges, advantages, and limitations associated with the preparation of these materials and to highlight their most recent energetic catalytic applications. The first section covers the structural properties of perovskites, including LaAlO_3_, and a general classification of these materials. 

The second section deals with different synthetic routes for some of the representative members of the LaAlO_3_ family, such as solid-state methods, solution-mediated methods (co-precipitation, sol–gel, and Pechini synthesis), thermal treatments (combustion, microwave, and freeze drying), and hydrothermal and solvothermal methods; these preparation methods are shown schematically in Figure 1. The third section shows the most recent energetic catalytic applications (DRM; SRM; the steam reforming of toluene, glycerol and ethanol; OCM; and TWC) using these functional materials (see Figure 1). Finally, the synthetic challenges, advantages, and limitations associated with the preparation methods and catalytic applications are also addressed.

## 2. Structure of the Perovskite

The ideal perovskite lattice (ABO_3_) is described as a cubic structure with *Pm3m* space group [22], where the A cations are in the corners in a 12-fold cub-octahedral coordination at atomistic position (½, ½, 0), while the B cations are in the center in six-fold coordination at atomistic position (½, ½, ½), and the oxygen atoms are situated at the face-centered positions [5]. Atom A in the cubic perovskite unit cell (see Figure 2) can be a metal of the IA or IIA group, a rare earth metal, or another large cation, such as K^+^, Na^+^, Ca^2+^, La^3+^, and Y^3+^. The B ions can be 3d, 4d, and 5d transition metal ions [7]. In the crystalline structure of perovskite, more than 90% of the elements of the periodic table can be incorporated [22]. The structure is depicted as a three-dimensional network of regular corner-linked BO_6_ octahedra [5]. The cations for ABO_3_-type perovskites commonly found in their structure are summarized in Table 1.

Two coordinations coexist in the same crystal attributed to the different ionic size between both cations (A and B). A is larger in size than B and similar to the anions, thus, favoring a more compact packing. However, as the volume of B is significantly lower than that of X, the most favored coordination is octahedral [23]. The perovskite structure is formed when it meets the criteria of the tolerance factor *t,* within the range (0.75 ≤ *t ≤* 1.0) (*r* is the ionic radius of the A, B) [5,7,22].
$$t=\frac{r_{A}+r_{o}}{\sqrt{2}\left(r_{a}+r_{o}\right)}$$

If the tolerance factor is 1, which adopts an ideal cubic structure, although this structure can be preserved with 0.75 ≤ *t* ≤ 1.0 [22,24]. If the value of *t* is less than 0.75 or between 1.00 and 1.13, the compound exhibits an hexagonal ilmenite structure (FeTiO_3_) or hexagonal symmetry, respectively [22]. Figure 3 shows the effects of the ionic size of A and B cations on the distortions observed in the crystal structure of perovskite for A^2+^B^4+^O_3_ and A^3+^B^3+^O_3_ combinations [25]. It can be seen that the value of *t* is inversely proportional to the distortions generated from the unit lattice. For this reason, for example, the triclinic system presents the lowest values of *t*, contrary to the cubic system [25].

Persovkite can display different phases depending on the temperature (T); thus, when T ≤ 100 °C, this material adopts an orthorhombic (γ) phase, whereas the tetragonal (β) phase starts to appear as the temperature increases to 130 °C, slowly replacing the previous (γ) phase [26,27]. As the temperature increases further to about 330 °C, the material starts to appear as a stable cubic (α) phase replacing the tetragonal (β) phase [28]. Those three crystal structures are included in Figure 4.

### 2.1. Structural Properties of LaAlO_3_ Perovskites

At room temperature, LaAlO_3_ perovskite has a rhombohedral structure with a R-3c space group. Figure 5a,b shows the primitive cell of LaAlO_3_ with two formula units (rhombohedral and cubic) with the respective lattice parameters (a), which represents the length of the three vectors and also the angles α = β = γ between these vectors. The crystal structure undergoes a transition from rhombohedral to cubic LAO above a temperature of 813 °C [29,30]. Due to their large band gaps, ternary cubic perovskite oxides exhibit a wide range of applications [29]. 

Moreover, if the conditions applied are varied, LAO exhibits numerous different structural, electronic, optical, elastic, and other properties. A few lanthanum-based oxides exhibit the cubic structure shown for LaAlO_3_ [29,31]. This structure of LaAlO_3_ is presented in Figure 6. La^3+^ is the cation placed in the corners (at site A, blue ball), and Al^3+^ is the cation placed in the center (at site B, black ball) [29,32]. Both ions are covalently bond with oxygen. Most LaAlO_3_ perovskites are pseudo-cubic or adopts phases with tetragonal, rhombohedral or orthorhombic crystal structures. LaAlO_3_ compounds have a critical temperature, Tc, of 500 °C with a transition from rhombohedral to a pseudo-cubic structure [29,32].

### 2.2. Classification of Perovskites

ABO_3_ perovskite-types are classified according to the radii of the metallic ions on the crystal structure [5,33] due to the flexibility and ability to incorporate a large amount of cations metallic with various oxidation numbers [5]. Thus, the main characteristic of these compounds is the possibility for several substitutions that are isomorphic at the A and B positions [5,33] leading to the production of large families of perovskites with different cations in the A site (A_x_A_1−x_BO_3_), with various cations in the B position (AB_x_B_1−x_O_3_) or substituted at both cation positions (A_x_A_1−x_B_x_B_1−x_O_3_) [5]. The oxidation states of both cations are close to 2+ (at site A) and 4+ (at site B), although both elements can be 3. The oxide phases are divided into two types (ternary oxides of the ABO_3_ type and newer compounds) [34,35], and the detailed classification is shown in Figure 7:Ternary oxides of the ABO_3_ type and their solid solutions: these can be classified as A^1+^B^5+^O_3_, A^2+^B^4+^O_3_, A^3+^B^3+^O_3_, and oxygen and cation deficient phases on the basis of their oxidation numbers [5,35].Newer complex of the type (AB′_x_B″_y_)O_3_, where B′ and B″ are two different metallic cations in different oxidation states [5,36].

The newer complex can be divided into four subgroups [5,37]:Perovskites with oxygen deficient phases, namely A(B′_x_B″_y_)O_3−z_.Compounds containing equal amounts of the two B cations, namely A(B′_0.5_B″_0.5_)O_3_.Perovskites containing twice as much of the lower valence state element as the higher valence state element, namely A(B′_0.33_B″_0.67_)O_3_.Compounds containing twice as much of the higher valence state element as the lower valence state element, namely A(B′_0.67_B″_0.33_)O_3_.

The different partial substitutions in A or B-site, and also the introduction of a crystalline lattice defect can generate several interesting applications and properties. Thus, the LaAlO_3_ perovskites in which the A-site is occupied by the La^3+^ and the B-site is occupied by Al^3+^ have attracted attention recently as catalysts for environmental and energetic applications [10]. In addition, they are considered to be good candidates for different applications, such as sensors, microwave dielectric resonators, high-frequency capacitors, and superconductors [13]. 

In this work review, we present recent studies with LaAlO_3_ perovskite doped at the A-site with rare-earth ions (such as Tb^3+^, Eu^3+^, Dy^3+^) and at the B-site with Ni^2+^, Co^2+^, Mn^2+^, Fe^3+^, Ga^2+^ or noble metal ions (such as Rh^3+^, Ru^3+^, Ir^3+^, Pd^2+^, and Pt^4+^), amongst others, which are prepared via various synthetic routes and used in several of the catalytic applications mentioned above (see Figure 1).

## 3. Synthesis Routes to LaAlO_3_ Perovskites

Since the first preparation of perovskites using the ceramic method in 1970, several methods for the synthesis of these materials have been developed [1,22,38,39].

### 3.1. Solid-State Method

The solid-state reaction, also known as mechanosynthesis or mechanical alloying, is the most conventional processing for the preparation of these materials. In this process, the mechanical energy of the system is generated by the impact between grinding media with particles of ground solid precursors and the collision between the particles of powder, thus, allowing chemical reactions at low temperatures close to room temperature [1]. 

The energy range of the process is from 0.1 to 100 MJ/kg depending on the milling parameters applied and the type of mill [1]. The resulting stresses due the collisions cause destruction of the ground solid precursors favor the crystalline lattice defects, particle-size reduction, vacancies, dislocations, etc. [40]. These reactions occur through a diffusion-controlled mechanism with the migration of the ions from the bulk to the interface between particles [41,42] as shown in Figure 8.

Perovskites have traditionally been synthesized via a solid-state reaction method, a simple process with high calcination temperature (about 1500 °C) that can produce large quantities of product and is cost effective, among other characteristics [43,44]. Several studies have been published regarding the synthesis of LaAlO_3_ perovskites using this method aimed at improving their textural properties for use in various applications. Thus, LaAlO_3_ perovskites were prepared by Zhang et al. [45] using the ceramic method through a reaction between lanthanum(III) oxide and transition alumina by grinding at room temperature (RT). 

The solid-state reaction proceeded as the grinding time increased and was complete after 120 min. The resulting products had a high specific surface area (S_BET_) (>10 m^2^/g). In other studies, aluminum- and lanthanum-based perovskites were prepared via the solid-state reaction method aiming to use the products generated in various applications. For instance, Fabian et al. [46] synthesized La_1−x_Ca_x_AlO_3−δ_ (x = 0.05–0.20) perovskite ceramics by mechanosynthesis and evaluated the optoelectronic properties of these materials. They found that substitution by Ca^2+^ increased both the total electrical conductivity by 2–3 orders of magnitude compared to bulk (LaAlO_3_) and also oxygen-ionic conductivity. 

The resulting nanoparticles (NPs) had a crystallite size in the range 11–34 nm. The materials were prepared using metallic oxides as precursors. For each preparation, a mixture of precursors oxides in the appropriate amount was ground in a ball mill (at 600 rpm) using different times (up to 30 min). The powder-to-ball weight ratio was 1:40. The obtained powders were pressed into several disks at 270 MPa until obtaining a pellet-type geometry. Finally, these were calcined at 1450 °C for 12 h in an air atmosphere (except LaAlO_3_, which was calcined at 1700 °C).

Su et al. [47] prepared a Ca^2+^/Mn^2+^-doped LaAlO_3_ perovskite via ceramic method and investigated the contribution of different contents of Ca^2+^ and Mn^2+^ on the infrared radiation properties of LaAlO_3_ perovskite. They found that when 50 mol% Mn^2+^ and 40 mol% Ca^2+^ were incorporated into perovskite, the average infrared emissivity in the wavelength range 1–22 μm was increased (up to 0.90), which was higher than for bulk LaAlO_3_ (0.71). 

The results of X-ray diffraction (XRD) demonstrated that the Ca^2+^/Mn^2+^-doped LaAlO_3_ perovskite obtained showed a trigonal rhombohedral crystal system with all peaks in accordance with characteristic diffraction peaks of the pure LaAlO_3_, which confirmed that the perovskite-type structure was obtained. La_2_O_3_ and Al_2_O_3_ were used to prepare LaAlO_3_ perovskite, whereas Ca^2+^/Mn^2+^-doped LaAlO_3_ was synthesized by mixing La_2_O_3_ and Al_2_O_3_ as the main oxide precursors together with CaCO_3_ and MnCO_3_ as doping agents. These solids reagents were mixed in ball-milling according to the molecular formulas of LaAlO_3_ and La_1−x_Ca_x_Al_1−y_Mn_y_O_3_ (0.2 ≤ x ≤ 0.4, 0.15 ≤ y ≤ 0.5) and then dried at 120 °C. The obtained powders were pressed into several cylinders at 120 and 200 MPa until obtaining a cylindrical-type geometry. Eventually these materials were calcined at 1550 °C for 2 h.

The mechanical behavior of ferroelastic lanthanum perovskite LaMO_3_ (M = Co (LCO), Al (LAO), Ga (LGO), and Fe (LFO-B and LFO-R)) under uniaxial compression was studied by Araki et al. [48]. The authors determined that the rhombohedral ferroelastic materials (LCO and LAO) exhibited a smaller stress range for domain switching than the orthorhombic ones (LFO-R, LFO-B and LGO). According to the rhombohedral angles obtained from their XRD patterns, the crystal structure of LaAlO_3_ perovskite was close to cubic. 

In addition, no other phases were founded in any materials obtained except for some traces of lanthanum(III) oxide in LGO. Perovskites were prepared by a solid-solution technique. Thus, metal oxide precursors were mixed with La_2_O_3_, were ground in a ball mill for 24 h using a medium of ethanol, and then calcined. The products obtained were ground in a ball mill again and then compressed into discs at 94 MPa for 5 min, followed by sintering. LaCoO_3_ and LaAlO_3_ were sintered twice.

Although a solid-state reaction is a conventional means of preparing LaAlO_3_ perovskites, this method requires high calcination temperatures (up to 1500 °C). As such, considerable research efforts have been made to synthesize these materials at lower calcination temperatures (700–1000 °C) [44,49,50,51,52]. It has been reported that the calcination temperature to obtain LaAlO_3_ perovskite can be reduced using orthorhombic (o-)LaCO_3_OH instead of La_2_O_3_ powders and can also be reduced by adding metal fluorides. It has been demonstrated that, during the formation of LaAlO_3_ perovskite, the transient species (δ and *h*-) La_2_O_3_ formed from thermal decomposition of o-LaCO_3_OH begins to disappear at lower calcination temperatures contrary to the addition of La_2_O_3_ powders. 

Fluoride ions play an important role in reducing the formation of LaAlO_3_ temperature; however, the reason why this occurs is under investigation. Although, it has been reported that the formation of transitory species (LaOF and LaF_3_) could be favorably affecting the process [44,53]. For instance, Lee et al. [44] prepared LaAlO_3_ by a ceramic method using the following precursors: La(NO_3_)_3_⋅6H_2_O, urea, NH_4_F, and alumina (Al_2_O_3_). These authors obtained a pure phase of perovskite lanthanum aluminate at 1000 °C—a much lower temperature than those in a conventional ceramic method. 

The XRD patterns of the powder prepared at different sintering temperatures (see Figure 9) show that the characteristic diffraction peaks of the pure LaAlO_3_ increased with increasing reaction temperature, as shown in Figure 9f, and that a single LaAlO_3_ phase was formed at 1000 °C. In order to prepare the perovskite, an aqueous solution of all precursors except Al_2_O_3_ was placed under reflux. Then, the solid product obtained (LaFCO_3_) was mixed with Al_2_O_3_ in an agate mortar, and subsequently this product was calcined between 700 and 1100 °C for 5 h under an air atmosphere.

The molten salt synthesis method is a novel solid-state preparation strategy that has attracted increasing attention in the past few years. Alkali metal salts are used in large quantities in this method, which is considered to be one of the simplest, versatile and most cost-effective methods for producing crystallized, chemically purified single-phase powders at low temperatures [43]. For instance, Mendoza et al. [51] prepared perovskite-type LaAlO_3_ at 350 °C using precursors, such as alkali metal nitrates (LiNO_3_, NaNO_3_, and KNO_3_). The experiment conducted by the researchers is shown in Figure 10.

LaAlO_3_ perovskites were prepared by Jin et al. [54] from a co-precipitation method of lanthanum(III) hydroxide and aluminum hydroxide, followed by a molten-salt synthesis method or solid-state reaction. Their procedure gave perovskite lanthanum aluminate highly pure at 1000 °C after 3 h in the presence of molten Na_2_SO_4_, as can be seen in the X-ray diffraction pattern (see Figure 11). 

To prepare the LaAlO_3_ perovskite, equimolar amounts of metal nitrate precursors were first dissolved in deionized water (DI) and stirred magnetically for 30 min. The reaction solution was then added slowly to a 2 mol/L aqueous ammonia solution with vigorous electromagnetic stirring. The obtained solid was washed with DI repeatedly and then dried at 120 °C for 24 h and sieved through a #200 mesh. The powder was homogeneously mixed with Na_2_SO_4_ in an agate mortar at a weight ratio of 3:1 (Na_2_SO_4_/(La_2_O_3_ + Al_2_O_3_)). The materials were calcined at different temperatures (850, 900, 950, and 1000 °C) for 3 h at a heating rate of 5 °C/min.

Recently, Beheshti et al. [13] published an innovative method to obtain LaAlO_3_ perovskites through a thermal shock assisted solid-state method. The LaAlO_3_ particles obtained using this method formed nanometer-sized crystals (~90 nm) with a rhombohedral structure. These materials were used for microwave absorption applications. The electromagnetic absorbability results indicated that the material based on LaAlO_3_ perovskites and paraffin showed excellent microwave absorption qualities. To prepare perovskite lanthanum aluminate, stoichiometric amounts of both powdered precursors (La_2_O_3_ and Al_2_O_3_) were mixed together directly. 

The resulting material was then heated to 900 °C at a heating rate of 10 °C/min. After this time, a thermal shock was applied by dropping the powder suddenly into liquid nitrogen. The resulting product was dried overnight at room temperature. Ethanol was then added to this material, and the product was agitated ultrasonically for half an hour and then heated at 80 °C. In the next step, the dried solid milled in a planetary ball mill (at 250 rpm) for 60 min. Finally, the material was calcined at 1300 °C for 60 min.

In general, the solid-state method uses nitrates, oxides, and carbonates for metal cations A and B in a stoichiometric ratio. All raw materials are mixed in ball-milling using a medium of isopropanol, ethanol, or acetone [55]. Then, the resulting product is dried and finally calcined at temperatures of up to 1100 °C [55,56]. This allows the formation of a pure phase of the perovskite structure [55]. 

However, the final material shows a high particle size (>1000 nm), broad particle distribution, low specific surface area (<2.5 m^2^/g), formation of other phases due to insolubility of the cationic species in the reaction medium [42,55], and the introduction of impurities during milling. Efforts continue to improve the solid-state technique with the aim of increasing the specific surface area, phase stabilization, and to purity the perovskites obtained. The characteristics of solid-state reaction methods for perovskite preparation as well as their advantages and limitations are shown in Table 2. This table also provides the characteristics of the particles synthesized (particle size and extent of agglomeration). The main characteristics (particle size and calcination temperature) of representative members of LaAlO_3_ family recently synthesized by solid-state and molten-salt reaction are also summarized in Table 3.

### 3.2. Solution-Mediated Methods

These methods include the co-precipitation of metal ions (using as precipitating agents oxalate, hydroxide, acetate, citrate, and cyanide complexes) [1], sol–gel and the Pechini synthesis, which result pure and homogenous materials with high specific surface areas. From the solvent removal, the methods can be classified as (i) precipitation and then centrifugation, filtration, etc., for the separation of the liquid and solid phases and (ii) thermal treatment, such as sublimation, combustion, and evaporation, for solvent removal [1].

#### Co-Precipitation Method

This method requires high levels of supersaturation in which the precipitation agent is mixed with solutions containing soluble metal cations [1]. The precipitated product is filtered and washed with DI. Finally, the resulting precipitate cake is dried and calcined. A general diagram of this process is shown in Figure 12. Several parameters, such as the pH, mixing rate, concentration, and temperature, must be taken in consideration when using this method to achieve the desired physical features (i.e., particle size distribution and morphology) [42]. 

Precipitation involves nucleation and subsequent growth in a solvent medium. The amount of nucleation sites determines the growth of uniform particles. Otherwise, by means of Ostwald ripening processes, growth continues to form large particles, which combine into aggregates with a broad particle-size distribution, morphology, shape and properties [60]. When the co-precipitation method is used to prepare perovskites, the shape of the structure obtained is determined by the crystal structure. 

**Table 3 materials-15-03288-t003:** Members of the LaAlO_3_ perovskite family recently synthetized using different methods.

Perovskite	Preparation Method	Particle Size (nm)	Calcination Temperature (°C)	Refs.
**La_1−x_Ca_x_AlO_3−δ_**	Mechanosynthesis	11–34	1450 and 1700	[46]
**Ca^2+^-Mn^2+^ doped LaAlO_3_**	Solid-state reaction	*	1550	[47]
**LaMO_3_ (M = Co, Al, Ga, Fe)**	Solid solution method	*	800–1500	[48]
**LaAlO_3_**	Solid-state reaction	*	1000	[44]
**LaAlO_3_**	Co-precipitation and molten salt synthesis	200–400	1000	[54]
**LaAlO_3_**	Thermal shock assisted solid-state method	90	1300	[13]
**LaAl_1−x_Ni_x_O_3−δ_**	Co-precipitation	*	700	[61]
**LaAlO_3_ doped with Pr (III) and Yb (III)**	Precipitation	100	1300	[62]
**LaAlO_3_**	Co-precipitation	*	950	[63]
**Eu^3+^-doped LaAlO_3_**	Co-precipitation	20	700 to 900	[64]
**Mg-substituted LaAlO_3_**	Citrate sol–gel	11.4–18.7	850	[65]
**LaAlO_3_ perovskite partially substituted with Ca or Ce**	Citrate sol–gel	*	800	[66]
**La_1−x_M_x_AlO_3−_** ** _δ_ ** **(M = Sr, Ba, Ca)**	Citrate sol–gel	*	850	[67]
**LaAlO_3_, La_0.7_M_0.3_AlO_3−δ_ (M = Sr, Ba, Mg, Ca), LaAl_0.7_M’_0.3_AlO_3−δ_ (M′ = Fe, Co, Mn, Ti, Cr), and La_1−x_Ca_x_AlO_3−δ_**	Citrate sol–gel	*	700	[68]
**LaAlO_3_**	Citrate sol–gel	*	950	[69]
**LaAlO_3_**	Sol–gel and modified Pechini	29–41	600–800	[8]
**LaAlO_3_:Bi^3+^, Tb^3+^**	Polyol mediated route	21	700	[70]
**LaAlO_3_ with 25% molar Al substitution by Co, Cu or Ga**	Citrate sol–gel	30	700	[71]
**Cu-doped LaAlO_3_**	Pechini-type sol–gel process	80–100	800	[15]
**Pd-substituted LaAlO_3_**	Citrate sol–gel	*	700	[72]
**Alkali-added** **LaAlO_3_ perovskite**	Citrate sol–gel	*	950	[73]
**(LaAlO_3_) and (RGO-LaAlO_3_)**	Gel route and low temperature combustion method	*	500	[74]
**Sr, Mn-doped LaAlO_3_ and Mn and Sr-codoped LaAlO_3_**	Pechini method	10–20	900	[21]
**LaAlO_3_:Ln^3+^ (Ln = Eu^3+^ or Tb^3+^)**	Pechini method	<60	900	[75]
**LaAlO_3_ perovskite**	Low-temperature solution combustion method	45	500	[14]
**Ionic substitutions of Pd, Pt, and Ru in LaAlO_3_ perovskite**	Combustion synthesis route	31–48	700	[18]
**LaAlO_3_:Eu^3+^**	Combustion synthesis route	70	600	[76]
**LaAlO_3_ perovskite**	Combustion synthesis procedure	36	1500	[77]
**Chromium-doped LaAlO_3_**	Combustion synthesis procedure	51–71	450	[77]
**Ni/LaAlO_3_**	Microwave assisted combustion	~41	900	[19]
**Ce/Mn dual-doped LaAlO_3_ perovskites**	Microwave sintering method	*	1400	[78]
**LaNi_x_Al_1−x_O_3_**	Hydrothermal method	*	800	[79]
**Dy^3+^/Eu^3+^ co-doped LaAlO_3_**	Hydrothermal technique	30	700	[80]
**RGO/LaAlO_3_**	Gel and hydrothermal methods	*	500	[81]
**LaBO_3_ (B: Mn, Co, Fe, Al and Ni)**	Supercritical hydrothermal method	*	450	[10]
**Eu^3+^ co-doped LaAlO_3_**	Thermovaporous method	100–700	400	[82]
**Eu^3+^ co-doped LaAlO_3_**	Solvothermal method	~90	800	[83]

* It was not reported.

For instance, the perovskite structure with a cubic symmetry adopts a cubic shape, trigonal or hexagonal symmetries lead to rods, and the orthorhombic structure gives spheres [60]. Some examples of co-precipitated perovskites with various crystal structures are presented in Figure 13. Various precipitating agents have recently been used to prepare LaAlO_3_, including oxalic acid, hydroxides, carbonates, aqueous ammonia, acetate, and cyanide complexes [42].

The methods based on oxalate involves the reaction of the oxalic acid with hydroxides, oxides, or carbonates, thus, generating metal oxalates, carbon dioxide, and water as products [1,7]. The solubility problems are minimized as the pH of the resulting solution is close to 7 [1,7]. 

Bělina et al. [84] prepared the mixed oxide LaNiO_3_ with a perovskite structure based on the decomposition of oxalate, which allowed the preparation of materials with an appropriate specific surface area, morphology, composition, and good catalytic performance. Similarly, Toprak et al. [85] synthesized nanostructured BSCF (Ba_x_Sr_1−x_Co_y_Fe_1−y_O_3_) by oxalate co-precipitation (see the general procedure in Figure 14). This method was shown to be effective for preparing the nanostructured BSCF with high purity and smaller particle size compared with conventional techniques.

The preparation methods based on hydroxides and carbonates can often be used due to the variety of precipitation schemes and low solubility [1]. Lanthanum-based perovskites are widely synthesized using hydroxide- and carbonate-based precipitation methods; for example, Djoudi et al. [61], using the co-precipitation method, synthesized the Ni-substituted perovskite lanthanum aluminate LaAl_1−x_Ni_x_O_3−δ_ (0 ≤ x ≤ 0.6). In this case, the aqueous solutions of metal nitrates with the desired Al/Ni ratio were mixed and precipitated with NaOH. 

After washing the solids with ethanol and DI several times and then drying overnight at 110 °C, the precursors obtained were ground into powders and heated to 700 °C for 6 h in an air atmosphere. Decomposition of the precursor hydroxides allowed the development of phase-pure perovskite with no detectable secondary phases. The morphology and microstructure of the products showed that the powders were partially agglomerated and that the sample particles were nearly spherical.

Varandili et al. [86] prepared B site co-doped LaFeO_3_ perovskite by co-precipitation using various alkaline agents, such as ammonium hydroxide and sodium hydroxide, and confirmed that the formation of a pure phase of perovskite with no impurities was achievable using NaOH or NH_4_OH. A reduction in particle size was observed in the samples co-precipitated with NH_4_OH compared with the powders co-precipitated with NaOH.

LaAlO_3_ nanocrystals doped with praseodymium(III) and ytterbium(III) via precipitation using aqueous ammonia as a precipitation agent were prepared by Lemański et al. [62]. The average size of the nanocrystallites formed was about 100 nm, and they were agglomerated into larger grains. In this study, several mechanisms for radiative and nonradiative energy transfer between the doped rare-earth ions were observed upon changing the excitation source, thus, meaning that these materials might be used as up- and down-converters, for example to enhance the yield of solar cells. 

Aluminum nitrate, lanthanum acetate, ytterbium acetate, and praseodymium chloride were used as starting precursors. The salts were dissolved in distilled water (DW) and then added to the precipitant solution (aqueous ammonia). The resulting suspension was then filtered by suction and washed several times with DW. The gel obtained was dried at 120 °C for several hours and then sintered at 1300 °C for 5 h. The amount of Pr^3+^ dopant in each sample was 1 mol%, and the amount of Yb^3+^ was 1, 3, and 5 mol%.

Recently, LaAlO_3__X catalysts were prepared by co-precipitation at several pH values (between 6 and 10) by Sim et al. [63]. The aim was to determine the effect of pH on the catalytic performance of perovskite lanthanum aluminate catalysts on the oxidative coupling of methane (OCM). LaAlO_3_ perovskite was successfully synthesized in catalysts LaAlO_3__7 (pH = 7) and LaAlO_3__8 (pH = 8), which had rod-shaped morphologies and specific surface areas of 2.1 and 2.5 m^2^/g. Among LaAlO_3_ catalysts prepared, the LaAlO_3__8 catalyst showed the highest C_2_ yield. 

Metallic precursors based on nitrate salts were used in the preparation, with sodium carbonate as precipitation agent. Thus, stoichiometric amounts of the nitrates salts precursors were dissolved in DW (Solution A), and Na_2_CO_3_ was dissolved in distilled water (1.5 mol/L) separately (Solution B). Both solutions were simultaneously added to 400 mL of DW at 70 °C with continuous stirring. In the co-precipitation process, the pH value was varied from 6 to 10 at intervals of 1 by adjusting the amount of Solution B. The solution obtained was stirred at 70 °C for 60 min, and then this was filtered. The resulting solid was washed with DW, dried overnight at 120 °C, and then ground and calcined in an air atmosphere at 950 °C for 5 h.

LaAlO_3_ has also been prepared using acetate complex. Wu et al. [64] prepared Eu^3+^-doped LaAlO_3_ perovskites with a Eu^3+^ concentration in the range 0–5 wt.%. These materials were successfully prepared employing acetic acid as the solvent. Crystalline LaAlO_3_ phases increased with temperature, and pure LaAlO_3_ was obtained at 900 °C. The average size of the samples was about 20 nm. During the preparation, Eu_2_O_3_ powders were dissolved in deionized water at about 80 °C, and then this solution was added to the previous homogeneous solutions to give a nominal Eu:La atomic ratio in the range of 0.5–5 wt.%. 

The solution obtained was stirred for 1 h at RT. The resulting product was then heated at 80 °C for several hours and dried at 120 °C before charring the organic contents at 350 °C for 1 h. Finally, the mixture was calcined in an air atmosphere at temperatures between 700 and 900 °C. Zhang et al. [87] synthesized La_1−x_Sr_x_Co_1−y_Fe_y_O_3_ using iron nitrate and acetate precursors for La, Sr, and Co and then calcining at 850 °C in an air atmosphere for between 5 and 10 h. According to the XRD data, all the oxides, except SrCoO_2.5_, showed a perovskite structure [7].

The cyanide complex method presents some advantages, such as low calcination temperature and control of stoichiometry. Asamoto et al. [88] prepared LnFe_x_Co_1−x_O_3_ (Ln = Gd, Er, Sm, La, Yb, Ho, Pr, and Dy) perovskites by the thermal decomposition of cyano metal complexes, which were heated at 300 °C for 1 h and then calcined in air at temperatures ranging from 500 to 1000 °C for 1 h. Pure phases of LnFeO_3_ perovskites were synthesized at the following calcination temperatures: 600 °C for Ln = Pr and La, 700 °C for Sm, 800 °C for Gd, 900 °C for Dy, and 1000 °C for Yb. The minimum calcination temperature required to obtain a pure phase of LnFeO_3_ decreased with decreasing atomic number of the lanthanoid. 

Sanchez-Rodriguez et al. [89] synthesized a LaFeO_3_ perovskite by solid-state thermal decomposition of a cyano metal complexes precursor. These authors prepared 1 mol/L aqueous solutions of La(NO_3_)_3_·6H_2_O and K_3_Fe(CN)_6_, and then these were mixed at RT while stirring constantly. After that, the mixture was filtered by suction, and the resulting solids were washed with DI, diethyl ether, and ethanol and subsequently dried in an air atmosphere at 50 °C. Then, the as-precipitated sample was moistened with ethanol and then milled. The average crystallite size and specific surface area of the final materials were varied from 27 to 40 nm and from 7 to 20 m^2^/g, respectively.

In general, co-precipitation methods allow control of the size and shape of perovskites and are also simple and environmental friendly. However, they lack an overall optimization due to the controls required during the washing step [42]. The characteristics of co-precipitation methods for perovskite preparation and their advantages and limitations are included in Table 2, which also lists the characteristics of the particles synthesized (particle size and extent of agglomeration). Finally, the main characteristics (particle size and calcination temperature) of representative LaAlO_3_ perovskites recently synthesized via the co-precipitation method are summarized in Table 3.

### 3.3. Sol–Gel Synthesis

Perovskites with high homogeneity, purity, and controlled composition can be prepared using the sol–gel method. This is a “bottom-up” process in which the particles or porous structures are assembled from elemental or molecular components [60]. The sol–gel technique has three important steps [60]: (i) mixing the precursor chemicals, namely a gel-forming medium (or a polymer), solvent, and catalyst; (ii) chemical treatment, which accelerates polymerization of the mixture to form the gel; and (iii) thermal treatment at high temperature to remove organic and volatile phases from the amorphous gel or xerogel to give a crystalline material with the required shape. As shown in Figure 15, several morphologies can be obtained depending on the hydrolysis and condensation performed to form the polymer.

The sol–gel modified Pechini method has become one of the most traditional synthetic methods for perovskites oxides due to its versatility and simplicity. Figure 16 shows the schematic representation of this synthesis method. For the preparation of perovskites, the metal ions (metal salt precursors—i.e., nitrates, acetates, etc.—in DW) are complexed using generally citric acid or ethylendiaminetetraacetic (EDTA), and ethylene glycol is added as stabilizing agent [60,90]. Other substances that can be used as chelating agents include tartaric acid [91], glucose [92], or oxalic acid [93]. 

The most common chelating agent used is citric acid due to its acid properties [94], in this case, the carboxylic acid group form the strongest complexes. The citric acid has three p*K*_a_ values (3.13, 4.76, and 6.39) [94]. Generally, this chelating agent is added in excess, normally with a citric acid:cation molar ratio of between 1 and 3. Normally, to promote polymerization is added ethylene glycol (polyalcohol) to form a polyester. The mixture obtained is then heated in the range 80–90 °C to form a viscous solution [94]. Finally the product obtained is calcined to obtain the perovskite [90,95,96].

The sol–gel modified Pechini method has been widely used to prepare LaAlO_3_. For instance, Bai et al. [65] synthesized Mg-substituted LaAlO_3_ perovskites using this method and used them as supports for Ni catalytic active sites in the CO_2_ reforming of methane. According to the X-ray diffraction results, the rhombohedral perovskite structure was formed, and no secondary peaks for Mg compounds were identified. 

The latter finding indicates that Mg^2+^ cations were incorporated into the perovskite lattice or were present as a non-crystalline phases. The specific surface area of resulting catalyst was low (<10 m^2^/g), as is typical for perovskite-based catalysts. Furthermore, these authors found that the Ni/La_0.8_Mg_0.2_AlO_2.9_ catalyst showed the highest catalytic performance. To prepare the LaAlO_3_ perovskite, a stoichiometric amount of the nitrates salts precursors were dissolved in 25 mL of DI, and then 10 mmol of citric acid was added to the above solution. 

The solution was heated at 55 °C with vigorous stirring to form a gel-like sample, which was further dried at 100 °C overnight. The solid was then calcined at 400 °C for 0.5 h and 850 °C for 7 h. The La_1−x_Mg_x_AlO_3−δ_ (x = 0.1, 0.2, 0.3, and 0.4) perovskites were synthesized following a similar method except that lanthanum(III) nitrate salt precursor was partially substituted with the equivalent amount of magnesium nitrate hexahydrate. Finally, Ni-perovskite catalysts with a Ni loading of 10.0 wt.% were prepared by the deposition-precipitation method.

Hernandez Martinez et al. [66] synthesized a LaAlO_3_ perovskite substituted with Ce or Ca and evaluated it in the ethanol SR reaction. The perovskites obtained presented a good crystallinity, and the diffraction lines were identified as those corresponding to the perovskite lanthanum aluminate phase. The S_BET_ for LaAlO_3_ perovskite was around 10 m^2^/g, increasing upon partial substitution of La with Ce (15 m^2^/g) and to a greater extent with Ca substitution, giving a value that was almost twice the original (22 m^2^/g). Furthermore, the inclusion of Ce promoted the mobility of active oxygen species on the catalyst surface. 

The Rh/LaAlO_3_ 0.3 wt.% catalyst showed the best catalytic performance with the highest mean value of H_2_ product distribution, low production of byproducts (methane and carbon monoxide), and long-term experience performance during 24 h on stream. The support perovskites were synthesized using the sol–gel modified Pechini method. Thus, the required amounts of metal nitrate precursors were dissolved in DI and mixed with an excess of citric acid. The resulting solution was then slowly evaporated at 65 °C under vacuum in a rotary evaporator until a gel was obtained. This product was dried in a vacuum oven overnight at 100 °C, ground, and calcined at 800 °C for 5 h in air at a heating rate of 10 °C/min to give three supports, designated as La_1−x−y_Ce_x_Ca_y_AlO_3_ (x = 0 when y = 0.1 and x = 0.05 when y = 0). Rh-based catalysts were obtained by the wet impregnation method.

Sato et al. [67] prepared La_1−x_M_x_AlO_3−δ_ (M = Ba, Sr, Ca; x = 0.0–0.3) materials and investigated their catalytic behavior for the OCM in an electric field at 150 °C. The La_1−x_Ca_x_AlO_3−δ_ catalysts prepared showed a perovskite structure, with specific surface areas in the range 5.2–20.5 m^2^/g. Among the catalysts evaluated, La_0.7_Ca_0.3_AlO_3−δ_ achieved the best performance. These perovskite oxide catalysts were synthesized as follows: First, metal nitrate precursors were dissolved in DW and stirred. Then, ethylene glycol and citric acid were then added to this solution (metal/citric acid/ethylene glycol molar ratio of 1:3:3). The resulting solution was then evaporated at 80 °C for 15 h until a gel was obtained. The product obtained was then precalcined at 400 °C for 2 h and then calcined at 850 °C for 10 h.

Various perovskites catalysts, including LaAlO_3_, La_0.7_M_0.3_AlO_3−δ_ (M = Ca, Mg, Ba, and Sr), LaAl_0.7_M’_0.3_AlO_3−δ_ (M’ = Cr, Ti, Fe, Co, and Mn), and La_1−x_Ca_x_AlO_3−δ_ (x = 0.1–0.5), were prepared by Yabe et al. [68]. These authors evaluated the catalytic behavior of these materials in the OCM at 150 °C in an electric field, using CO_2_ as an oxidant. La_0.7_Ca_0.3_AlO_3−δ_ showed high CO_2_-OCM activity. Ca-doped or supported LaAlO_3_ perovskite showed specific surface areas of between 4.0 and 12.3 m^2^/g. These materials were prepared using the same procedure reported by Sato et al. [67]. Several methods to prepare three LaAlO_3_ perovskite were considered by Lee et al. [69], i.e., citrate sol–gel (LaAlO_3__C), solid-state (LaAlO_3__S), and hard-templating (LaAlO_3__H). These authors demonstrated that the relative crystallinity of the resulting LaAlO_3_ materials differed depending on the synthetic procedure, thereby, directly affecting their catalytic performance. 

Thus, the catalyst synthesized using the sol–gel modified Pechini method (LaAlO_3__C) showed high relative crystallinity (100%), with an S_BET_ of around 3 m^2^/g, and showed the highest catalytic performance. The detailed schematic representation of the preparation methods are shown in Figure 17. In the case of the modified Pechini method, 2.7 g of lanthanum nitrate and 4.8 g of citric acid were dissolved in 75.0 mL of DI (Solution A) and 2.3 g of aluminum nitrate was separately dissolved in 25.0 mL of DI (Solution B). Then, the solution B was added dropwise to the solution A, with vigorous stirring. After stirring the mixed at RT for 1 h, the resulting solution was evaporated until a gel was obtained, then dried overnight at 200 °C, ground, and calcined at 950 °C for 5 h in air.

Silveira et al. [8] synthesized lanthanum aluminate (LaAlO_3_) perovskites by four routes: sol–gel with copaiba oil, sol–gel with coconut oil, sol–gel with coconut water, and modified Pechini (used as a reference). LaAlO_3_ perovskites with a single-crystalline phase (space group R-3c) were prepared at 600 °C using the sol–gel with coconut oil method. In the samples synthesized using the reference method, this phase was formed at around 700 °C (800 °C for the other routes). The preparation methods with natural reagents gave solids with smaller particles sizes (29–37 nm) than those prepared for the reference method (41 nm). 

The materials obtained presented various morphological characteristics, such as appearance of small compacted grains, small blocks, and a sponge-like appearance. The preparation with natural reagents may leave traces of contaminants in the prepared solid; however, they promoted the production LaAlO_3_ perovskites at lower temperatures than those employed in conventional procedures. The sol–gel synthesis of perovskites was conducted using different natural complexing agents. In this study, samples produced in the presence of coconut water were named SGC, those with coconut oil were named SGL, and those with copaiba oil were named SGO. 

Initially, La_2_O_3_ was dissolved in an aqueous solution of HNO_3_. To prepare SGC samples, nonahydrate aluminum nitrate was dissolved in 50 mL of coconut water, mixed with the La^3+^ ion solution, after drying at 120 °C for 24 h. After drying, the solid was ground and homogenized. To prepare SGL samples, aluminum nitrate was dissolved in 15 mL of ethanol and mixed with the La^3+^ ion solution (Solution A). Separately, coconut oil was dissolved in ethanol at a 1:1 oil-to-alcohol ratio (Solution B), and then the solution A was added to the solution B. 

The material obtained was heated to 250 °C for 3 h to evaporate solvents. SGO materials were prepared similarly to SGL materials, replacing coconut oil with copaiba oil. All resulting products from these synthesis were calcined at 500 °C for 4 h. The resulting powders were then ground, homogenized, and subdivided into three batches, which were calcined at 600, 700, or 800 °C.

Recently, Shaik et al. [70] synthesized LaAlO_3_: Bi^3+^, Tb^3+^ nanoparticles (LNP), and then these were surface modified with chitosan (CS) for targeted drug delivery and bioimaging of breast cancer cells. These authors used capecitabine (CAP) as a model drug to assess the drug delivery ability of LNP. The LaAlO_3_ perovskite obtained had a rhombohedral crystal system (space group R3m). The resulting nanoparticles had a crystallite size of 21 nm, with a spherical morphology. 

The results obtained also showed that these functionalized nanoparticles exhibited greater cellular internalization and significant cytotoxicity (IC_50_ = 9 μg/mL) when compared with naïve LNP and releases the drug specifically at the pH of the tumor microenvironment. LaAlO_3_:Bi^3+^ (1 wt.%), x% Tb^3+^ (x = 0, 0.5, 1, and 2) nanoparticles were prepared by a simple polyol-mediated route. First, metal nitrates were dissolved in 10 mL of DW. This mixture was then added to 20 mL of ethylene glycol. 

The resulting solution was stirred and heated at 100 °C, and then urea (2 g) was added followed by heating at 200 °C. The resulting solid was centrifuged, washed with methanol and acetone, and this was dried overnight at RT. The product obtained was then calcined at 700 °C for 5 h. The schematic representation of the synthesis of these functionalized nanoparticles and their cellular internalization mechanism is presented in Figure 18.

Tran et al. [71] prepared Cu/Co/Ga-doped LaAlO_3_ catalysts using the sol–gel method and evaluated them in the gas-phase conversion of ethanol. The XRD patterns of the catalysts showed that they contained a single perovskite phase. The S_BET_ was in the range between 11 and 16 m^2^/g, and the crystallite size was around 30 nm for all samples. In the case of perovskite lanthanum aluminate, partial substitution of Al significantly affected the catalytic activity and selectivity. For preparation, stoichiometric amounts of the nitrate salts precursors were dissolved in DI. Then, citric acid (citric acid/metal molar ratio of 2:1) was added to the above solution at pH = 7.5. The resulting solution was then evaporated at 80 °C for until a gel was obtained. The gel was heated at 150 °C for 3 h and calcined at 700 °C for 5 h in an air atmosphere.

Wang et al. [15] synthesized a Cu-doped LaAlO_3_ perovskite (LaAl_1−x_Cu_x_O_3_) (x = 0, 0.01, 0.02, 0.05 and 0.1), using sol–gel modified Pechini method, for the oxidative degradation of organic pollutants (phenol, diphenhydramine, ciprofloxacin, ibuprofen, phenytoin, 2-chlorophenol, 2,4-dichlorophenoxyacetic acid, and bisphenol A). The experimental results showed a high dispersion of copper species in the perovskite lanthanum aluminate. Furthermore, LaAl_1−x_Cu_x_O_3_ perovskites exhibited a spherical morphology, with an average crystallite size in the range of 80 to 100 nm. 

The catalyst LaAl_0.95_Cu_0.05_O_3_ allowed an efficient degradation of organic pollutants and was stable under heterogeneous Fenton-like conditions at a pH of 5.0–9.0. Catalysts were prepared via sol–gel modified Pechini method. Thus, to prepare Cu-doped LaAlO_3_ perovskites, the required amounts of metal nitrate precursors were dissolved in 20 mL of solvent (ethanol:deionized water = 7:1). PEG and citric acid and were then added to the above solution at a citric acid to metal ion molar ratio of 2:1. The resulting solutions were evaporated at 75 °C in a water bath for 6 h until a gel was obtained and then dried at 110 °C overnight. The solid products were well ground and precalcined at 450 °C for 4 h. Finally, the materials were calcined at 800 °C for 3 h.

Yoon et al. [72] synthesized a Pd-substituted perovskite lanthanum aluminate catalyst by the citric acid method, finding that Pd was incorporated into the lattice structure of supports (LaAl_1−y_Pd_y_O_3_, y: 0.016, 0.032 and 0.049 equivalent to Pd loadings of 0.8, 1.6 and 2.4 wt.%, respectively). The S_BET_ of the resulting LaAlO_3_-based catalysts was in the range of 4.8 and 11.6 m^2^/g, with an average crystallite size of 37–74 nm. These catalysts exhibited high TWC activity and strong thermal stability. The La–Pd interaction in LaAlPdO_3_ catalyst was the primary cause of the high turnover frequency (TOF) and low light-off temperature (LOT) for the conversions of CO, C_3_H_6_, and NO. 

To prepare LaAlO_3_, the metal nitrate precursors were dissolved in distilled water and mixed with citric acid (10% excess with respect to the ionic equivalence of the cations). This solution was continuously stirred for 1 h at RT then evaporated at 80 °C. The resulting gel was heated at 110 °C overnight and pre-calcined at 270 °C. Finally, the catalyst obtained was calcined at 700 °C for 5 h in a muffle furnace. Palladium (0.8 wt.%) was impregnated onto LaAlO_3_ using the incipient wetness method, then dried overnight at 110 °C, and finally calcined at 700 °C for 5 h.

An et al. [73] prepared LaAlO_3__XY (X = Li, Na, K, Y = mol%) catalysts by the citrate sol–gel method and applied them to the OCM reaction. The S_BET_ of the materials synthesized was between 1.4 and 3.5 m^2^/g. Furthermore, XRD measurements confirmed that catalysts LaAlO_3__X5, LaAlO_3__X10, and LaAlO_3__X20 were successfully obtained, whereas LaAlO_3__X30 showed other peaks with the presence of LaAlO_3_ perovskite peak. 

To prepare these catalysts, stoichiometric amounts of La(NO_3_)_3_ and Al(NO_3_)_3_ were dissolved in 25 mL of DI separately (solution A and solution B, respectively). Then, 5 mol of alkali metal salt and 0.02632 mol of citric acid were dissolved in 100 mL of DI (solution C). After, solutions A and B were then added to solution C. Then, the resulting mixture was stirred at RT for 1 h, and it was evaporated at 80 °C. The product was heated at 200 °C, and the material obtained was ground and calcined at 950 °C for 5 h.

LaAlO_3_ and lanthanum aluminate supported in reduced graphene oxide (RGO-LaAlO_3_) as adsorbents for the removal of methyl orange (MO) were obtained by Alrobei et al. [74]. The S_BET_ areas of the RGO and RGO-LaAlO_3_ nanocomposite were 112 and 250 m^2^/g, and the maximum adsorption capacity was found to be 469.7 and 702.2 mg/g for LaAlO_3_ and RGO-LaAlO_3_. The authors prepared LaAlO_3_ perovskite via the gel route. To prepare this perovskite, a stoichiometric amount of lanthanum(III) oxide powder was dissolved in the 1:1 nitrating mixture and heated until gel formation. 

Stoichiometric amounts of aluminum nitrate and oxalyl dihydroazole were then added to the gel, which was heated to around 300 °C and maintained at that temperature. The RGO-LaAlO_3_ nanocomposite was prepared in a hydrothermal autoclave at 120 °C for 5 h.

The Pechini method was used by Da Silva et al. [21] to prepare Sr-doped, Mn-doped, and (Mn + Sr)-co-doped LaAlO_3_ perovskites. This method was designed to obtain materials that meet the requirements of high surface area, control of the particle size and stoichiometry, and a morphology that allows it to be processed as the anode for the OCM in solid oxide fuel cells (SOFC). These materials were successfully prepared using the Pechini method. According to the results, all materials showed a pure single phase of perovskite lanthanum aluminate. 

The specific surface area was in the range between 5.39 and 16.24 m^2^/g, and the average particle size was in the range of 10 to 20 nm with a roughly spherical morphology. Further, the Mn-doped and (Mn + Sr)-co-doped electrocatalysts showed a high electrical conductivity (<1 S/cm) for SOFC applications. To prepare these materials, ethylene glycol and citric acid were dissolved in DI. Equimolar amounts of metal nitrate precursors were then added (an ethylene glycol:citric acid:total metal ions molar ratio of 10:5:1 was used). The resulting solution was heated to 80 °C while stirring constantly. Finally, the resulting powder was ground and calcined at 900 °C for 6 h in an air atmosphere.

Garcia et al. [75] synthesized Ln^3+^-doped LaAlO_3_ (Ln = Tb^3+^ or Eu^3+^) with a Ln^3+^ loading in the range 1–9 at.%. LaAlO_3_ phosphors doped with Eu^3+^- or Tb^3+^ were sub-micrometer-sized, highly crystalline and monophasic, and had a crystallite size of less than 60 nm. In light of the optical performance observed for these materials, the authors determined that these solids may be use as luminescent nanophosphors. Pechini’s method was used for the syntheses. Thus, a terbium or europium nitrate salt precursor was prepared by dissolving the lanthanide oxide in nitric acid. 

Nonahydrate aluminum nitrate and lanthanum(III) oxide were then dissolved in HNO_3_, and a stoichiometric amount of europium (or terbium) nitrate solution was added. After 10 min, citric acid (2:1 citric acid:metal molar ratio) was added, the solution was heated to 80 °C (10 min), and then sorbitol (3:1 sorbitol:metal molar ratio) was added. The solution was stirred until a polymeric resin had formed. This resin was heated at 350 °C for 2 h. The product obtained was finally calcined at 900 °C for 3 h under an air atmosphere for Eu^3+^-doped and a CO atmosphere for Tb^3+^-doped materials.

Rivera-Montalvo et al. [97] prepared europium-, praseodymium-, and dysprosium-doped LaAlO_3_ perovskites for thermoluminescent applications using the Pechini synthesis. The results of this study showed that a single phase of perovskite lanthanum aluminate phase was prepared, with an average particle size of 1.6 μm, a rhombohedric structure and spherical agglomerates. 

Dy- and Pr-doped LaAlO_3_ showed a high intensity luminescence when excited by X-rays, thus, suggesting that these materials could be used as TL phosphors to determine the absorbed dose in radiotherapy treatment. These materials were prepared using nitrate salts as metallic precursors, ethylene glycol, and citric acid. Thus, after heating the resulting solution at 80 °C to trigger the polyesterification reaction, the powders obtained were calcined from 700 up to 1600 °C. The flow chart for the preparation of these materials is summarized in Figure 19.

Morales Hernandez et al. [11] synthesized Pr^3+^- doped LaAlO_3_ using the Pechini synthesis and spray drying. The XRD spectra for Pr^3+^- doped LaAlO_3_ materials confirmed the presence of a rhombohedral crystal system only. The morphology of the materials obtained indicated that they comprised porous agglomerates with a semispherical morphology (these particles had a crystallite size of 2 µm). The TL results showed that Pr^3+^- doped LaAlO_3_ could be useful in UVC radiation dosimetry applications (in the range of 100–290 nm) using the TL method. The materials were prepared using the methodology reported by Rivera-Montalvo et al. [97].

In general, the sol–gel and Pechini synthesis allow to prepare perovskites with high homogeneity and purity and also excellent control of the composition of the final product. However, these methods require high temperatures and long periods of time. The characteristics of the sol–gel and Pechini methods for perovskite preparation, and their advantages and limitations are summarized in Table 2. The main characteristics (particle size and calcination temperature) of representative LaAlO_3_ perovskites recently synthesized using these methods are also summarized in Table 3.

### 3.4. Thermal Treatments

Combustion represents another viable method for preparing perovskite oxides. The reaction is exothermic, self-sustaining, and can occur rapidly at lower operating temperatures. These conditions can be achieved using metal nitrates as precursors (oxidizing agents) and organic fuels (citric acid, glycine, urea, etc.). The organic fuel molecules play a twofold role: they avoid the precipitation of new species and, at the same time, they form complexes with metal cation precursors to improve the homogeneity [60]. A representative scheme is shown in Figure 20. 

Recently, Manjunatha et al. [14] synthesized a LaAlO_3_ perovskite using a low-temperature solution combustion method. This nano-adsorbent was evaluated for the adsorptive removal of a dye solution (Direct Blue 53 (DB-53)) and fluoride, and the authors also studied the in vitro antimicrobial effect against test microorganisms: two Gram-negative bacteria (*Escherichia coli* and *Pseudomonas aeruginosa*) and two Gram-positive bacteria (*Staphylococcus aureus*) and *Bacillus subtilis*). 

The maximum removal capacity of the perovskite was found to be 40.8 mg/g (fluoride) and 71.4 mg/g (DB-53), and it showed a maximum antibacterial effect with an minimum inhibitory concentration of 63 μg/mL for Gram-negative bacteria (*Pseudomonas aeruginosa*). The LaAlO_3_ perovskite obtained had the following characteristics: good crystal integrity, an average size of the nanocrystallites of 45 nm, and a S_BET_ of 5.91 m^2^/g. To prepare LaAlO_3_, 3.26 g of La_2_O_3_ was nitrated with 10 mL of mixture of concentrated nitric acid and sulfuric acid in a 1:1 ratio in a sand bath to form a gel. Aluminum nitrate (7.50 g) and oxalyl dihydrazide (7.08 g) were then added to this gel. The mixture was dissolved in 20 mL of DI and calcined at 500 °C for 5 min to give a white luminous powder.

Anil et al. [18] synthesized LaAlO_3_:Ru^3+^, Pt^4+^, and Pd^2+^ catalysts using a combustion method. The catalytic performance of these materials was examined for the DRM reaction. The S_BET_ of the perovskites obtained was between 4.0 and 16.0 m^2^/g, the average particle size was in the range of 31 and 48 nm, and the LaAl_0.98_Ru_0.02_O_3−δ_ catalyst showed the highest catalytic performance. Lanthanum nitrate, aluminum nitrate, ruthenium chloride, chloroplatinic acid, palladium, and glycine as fuel were used as precursors to prepare the materials. Thus, stoichiometric ratios of these reagents were added to 30 mL of DI and sonicated for 5 min. The solution obtained was heated in a muffle furnace preheated to 400 °C. The resulting material was ground and calcined at 700 °C for 3 h.

LaAlO_3_:Eu^3+^ was synthesized by Fu et al. [76] from a urea-assisted solution combustion method using metal nitrate precursors as oxidizing agents. According to the XRD patterns, the LaAlO_3_ materials showed a pure rhombohedral phase. These authors found that the average size of the nanocrystallites was approximately 70 nm. The morphology of the La_1−x_AlO_3_:xEu crystals was irregular spherical-like. These materials showed interesting optical properties, such as the emission of strong red luminescence and efficient luminescence under the excitation of ultraviolet radiation, which may make them excellent candidates for the manufacture of LEDs and the advancement of new optical systems. 

A series of La_1−x_AlO_3_:xEu (x = 0.03–0.21) materials were synthesized. Thus, europium(III) oxide was first dissolved in HNO_3_ to form an aqueous europium(III) nitrate solution (0.01 mol/L). Then, the appropriate amounts of above solution and CO(NH_2_)_2_ were mixed according to the required molar ratio. This mixture was then stirred for a few minutes and heated to 600 °C. Finally, the material obtained was ground.

Figueredo et al. [19] synthesized Ni/LaAlO_3_ powders by microwave-assisted combustion, and the resulting catalyst was studied for the DRM. The XRD spectra obtained showed that the LaAlO_3_ support synthesized had characteristic diffraction peaks of this perovskite with a rhombohedral crystal system and an average particle size about 40.77 nm. The results indicated that this Ni/LaAlO_3_ catalyst was more stable than Ni/α-Al_2_O_3_ (comparison catalyst) in the conversions of carbon dioxide and methane. 

To synthesize this compound, 10 mL of DW was mixed with stoichiometric amounts of the nitrate salt precursors. This mixture was stirred constantly at 60 °C. Then, 4.00 g of urea was added to above solution. The beaker containing the product was then microwave irradiated at an output power of 900 W and a frequency of 2.45 GHz in the range 3–5 min. Finally, the resulting solid was calcined at 900 °C for 2 h in an air atmosphere.

A pure phase of perovskite lanthanum aluminate by combustion method, using a β-alanine and urea fuel mixture was prepared by Ianoş et al. [77]. The LaAlO_3_ perovskite obtained had an average particle size about 46 nm and a S_BET_ of 3.0 m^2^/g. The addition of sodium chloride to the reaction medium decreased the combustion temperature and resulted in an almost tripling of the S_BET_ to 8.5 m^2^/g and a decrease in the average crystallite size to 36 nm. This perovskite was prepared using the required amounts of metal nitrates precursors and a single fuel (urea or β-alanine) or fuel mixture (urea and β-alanine). 

These authors also prepared a material using the required amounts of metal nitrate precursors, fuel mixture and sodium chloride (50 wt.% with respect to the theoretical yield of perovskite lanthanum aluminate). Thus, the required amounts of metal nitrates precursors (and sodium chloride) were dissolved in 20.0 mL of DW, and then the appropriate amount of fuel was added. The resulting clear solution was heated rapidly to 400 °C. The resulting products were ground, washed with DW, and finally dried. A flowchart for LaAlO_3_ preparation, sintering, and characterization can be found in Figure 21. 

The same authors [98] synthesized the Cr^3+^-doped LaAlO_3_ (LaAl_1−x_Cr_x_O_3_ (x = 0 and 0.05)), a pink-red pigment using a combustion reaction. The resulting nanoparticles (LaAl_1−x_Cr_x_O_3_) had a crystallite size in the range 51–71 nm and S_BET_ between 3.0 and 7.3 m^2^/g. Pigment testing showed that this compound can be used to prepare water-based acrylic paints. The materials were prepared as follows: First, metal nitrates precursors solids were heated at 120 °C, and stoichiometric ratios of glycine and urea were then added to the above product. In the case of one material, 3.2 wt.% of mineralizer (calcium fluoride) was added to the precursor solution, and then the obtained products were calcined at 450 °C. The obtained pigments were milled, washed with hot DW (80 °C), and heated at 120 °C for 12 h.

In general, the combustion method is applied to prepare highly pure, homogenous, and crystalline nanomaterials at low temperature with less heat requirements and lower costs. This method involves the combustion of metals precursors in aqueous solution with one or the mixture of organic fuels to obtain a fluffy material with a high surface area and volume [55]. The main limitation of this method is that large amounts of carbon are often produced as an unwanted by-product [55]. Furthermore, it is not easy to control the reaction temperature, which is associated with the nitrate-to-fuel ratio and fuel types. 

Low crystallization and impurities could also be present after the combustion reactions, and subsequent sintering is generally required in these cases [43]. The characteristics of combustion methods for perovskite preparation, and their advantages and limitations, are included in Table 2. The main characteristics (particle size and calcination temperature) of representative LaAlO_3_ perovskites recently synthesized using combustion processes are also included in Table 3.

Another method used to synthesize perovskite is the microwave-assisted technique, which has been used since the early 21st century to replace conventional heating [22]. Microwave methods employ high-frequency electric fields to heat the materials via dipolar polarization, ionic conduction, and dipole rotation. When polar molecules interact with the alternating electric field of the microwave, they move to align themselves with the electric field of the electromagnetic wave, which results in rotational motion of the molecules and the generation of heat as a result of intermolecular interactions. 

Recently, several LaAlO_3_ perovskites with high purity have been obtained using microwave heating. For instance, Kasala et al. [99] prepared a rare-earth aluminate (ReAlO_3_, Re = La, Y, Nd, and Gd) from a mixture of Al_2_O_3_ and Re_2_O_3_ (Re = La, Gd, Nd, and Y) solids using the microwave-assisted method. The LaAlO_3_ perovskite obtained had the following characteristics: good crystal integrity, rhombohedral crystal structure, 10% of all particles had a diameter of 20–45 µm and 65% a diameter between 1.73 and 19.90 µm, a specific surface area of 1.12 m^2^/g and, finally, a sphere-like morphology with micro grains.

According to these authors, the preparation method was environmentally friendly and inexpensive and thus appropriate for scale-up preparation. In the synthesis, Al_2_O_3_ and the metal oxide precursors (Re_2_O_3_) were mixed in a 1:1 molar ratio and microwave heated at 900 °C for 20 min. The resulting solids were ground in a ball mill with DI for 8 h using alumina balls. The suspension was then dried at 110 °C for 24 h. Finally, the materials were calcined using microwave heating at 1050 °C for 3 min.

Deng et al. [78] synthesized Ce and Mn-doped LaAlO_3_ perovskites using microwave heating. The perovskite-type LaAlO_3_ compounds were obtained with high crystallinity. LaAlO_3_ and Ce- and Mn-doped LaAlO_3_ materials had an irregular polygonal-like morphology. These authors showed that the far-infrared emissivity was effectively enhanced as a result of Mn- and Ce-doping, reaching as high as 0.932. The researchers used the methodology shown in Figure 22. 

To prepare La_1−x_Ce_x_Al_1−y_Mn_y_O_3_ (x = 0.2 and 0.4; and y = 0.15, 0.3, and 0.45) perovskites, lanthanum oxide, aluminum oxide, and manganese dioxide were mixed in the appropriate stoichiometric ratio in ball-milling for 4 h using a medium of ethanol (3%). The resulting product was screened, dried at 350 °C for 6 h, and then compressed into cylinders at 50 MPa using polyvinyl alcohol (PVA). These cylindrical pellets were dried at 100 °C for 2 h and finally calcined using microwave heating at 1400 °C for 4 h.

Microwave-assisted technique can be incorporated into other synthetic processes, such as sol–gel. Gonjal et al. [100] prepared the series of LaMO_3_ (M = Al, Cr, Mn, Fe, and Co) perovskites by microwave heating of the metal nitrate precursors. For doped solids, the technique has to be combined with the sol–gel method. These authors synthesized well-crystallized and pure materials and determined that this method is an interesting alternative. 

The LaAlO_3_ perovskite crystallized in the R-3c space group. During preparation, equimolar amounts of metal nitrates precursors were mixed with carbon black (5 wt.%) and then homogenized and compressed into 13-mm-diameter pellets, which were subsequently microwave irradiated at a frequency of 2.45 GHz and power of 800 W. Then, these materials were calcined in a conventional furnace at 1000 °C in an air atmosphere.

In general, microwave-assisted methods offer several advantages, namely (i) short preparation time, (ii) clean process with limited use of solvents for the preparation of functional materials, and (iii) high energy efficiency [22]. The limitations of this method are that it may be unsuitable for upscaling, the need for expensive equipment, and a lack of reaction monitoring.

### 3.5. Hydrothermal/Solvothermal Synthesis

A hydrothermal or solvothermal reaction is any heterogeneous phase chemical reaction at high pressure and temperature that occurring in the presence of solvent medium to crystallize and obtained materials directly from the solutions [60]. The term “hydrothermal” is generally used if the solvent is water and “solvothermal” if the reaction is conducted using any other solvent [60]. These methods are used for the preparation of uniform and fine particles with high crystallinity and purity [10]. During hydrothermal reactions, the precursors used are exposed to high temperature (near the supercritical point of water) and high pressure (e.g., 14 MPa) in an autoclave reactor [55,60]. 

The hydrothermal method involves an interaction between various parameters, such as the temperature, pressure, and solvent type, that govern the chemical reactions by controlling ionic mobility [60]. In the hydrothermal synthesis, first stoichiometric quantities of the precursor salts are mixed with a suitable solvent medium to form a sol, is depicted in Figure 23 [55,60]. Then, the resulting solution is sealed in an autoclave reactor at the temperatures of interest and pressure for the appropriate time, and then the autoclave is cooled to room temperature. Finally, the solution is ultrasonicated and the resulting solid is centrifuged and dried [55].

According to Luo et al. [60], the hydrothermal synthesis of perovskite oxides can be classified into two methods: (i) reagents and a mineralization agent are dissolved in water, and then the solution is heated to above the supercritical temperature of the water (>350 °C); (ii) autoclave vessels lined with Teflon are used; therefore, the temperature conditions for using such reactors are limited to 200 °C and pressures of less than 10 Mpa. Sagar et al. [79] prepared two series of LaNi_x_Al_1−x_O_3_ perovskites (0 ≤ x ≤ 1) using hydrothermal and sol–gel methods and then studied and compared the performance of these catalysts for DRM under similar conditions. 

According to the results obtained, the perovskite-type phase was synthesized with 0.2 < x < 0.8 using both synthesis methods. Furthermore, these compounds showed specific surfaces areas of between 1.9 and 18.6 m^2^/g. However, the catalytic performance of the best catalyst prepared by the hydrothermal method was better than that of the best catalyst prepared by the sol–gel method. During preparation, equimolar amounts of metal nitrates were separately dissolved in hot propionic acid. 

The resulting solutions were added and mixed in the following order: (i) nickel and aluminum propionate solutions and (ii) lanthanum propionate solution. Subsequently, the solution was stirred for 30 min and then kept under reflux for 24 h. This solution was poured into an autoclave reactor and heated at 150 °C for 24 h. The resin obtained was dried and the resulting solid was calcined at 800 °C with a temperature.

Ankoji et al. [80] synthesized Eu^3+^/Dy^3+^ co-doped LaAlO_3_ perovskites using the hydrothermal technique. According to the XRD patterns, the materials showed a pure perovskite-type phase with rhombohedral structure. These authors found that the average size of the polygonal particles was approximately 30 nm. They were used to prepare nanophosphors, which may be useful for warm white-light emitting materials. Thus, to synthesize La_1−x_AlO_3_ (x mol% Dy^3+^ (x = 3–12)) and La_0.91−y_AlO_3_ (9 mol% Dy^3+^, y mol% Eu^3+^ (y = 1–5)) nanophosphors, La_2_O_3_, Dy_2_O_3_ and Eu_2_O_3_ were transformed into nitrates, respectively, in the appropriate molar ratios by addition of HNO_3_ (concentrated). 

The resulting solution was heated, and then 20 mL of DW were added. Al(NO_3_)_3_⋅6H_2_O was separately dissolved in 20 mL of DW. Both resulting solutions were then mixed, and then 1 mol of citric acid was added with stirring. The reaction medium with a pH of 7 was maintained by dropwise addition of a NaOH solution. Subsequently, the resulting solution was stirred for 30 min, then poured into an autoclave reactor, and heated at 473 K for 24 h. The resulting solid was cleaned and washed with DW and acetone. The resulting product obtained was then dehydrated at 70 °C for 5 h. Finally, all samples were calcined at 700 °C for 2 h in air.

Rag et al. [81] synthesized LaAlO_3_ perovskite particles embedded with green-reduced graphene oxide (RGO-LaAlO_3_) using the hydrothermal synthesis. The resulting LaAlO_3_ and RGO/LaAlO_3_ powders were crystalline in nature with a perforated cage-like morphology. Furthermore, the specific capacitance of the best composite was high (721 F/g at a scan rate of 2 mV/s) compared to others reported previously in the literature. Thus, the RGO-LaAlO_3_ may be useful in energy-storage devices. It is worth noting that LaAlO_3_ was prepared by the gel route and the RGO/LaAlO_3_ composite material by a hydrothermal method. 

To prepare LaAlO_3_ perovskites, the authors dissolved a stoichiometric amount of La_2_O_3_ powder in a 1:1 nitrating mixture and then heated this mixture until a gel was formed. Stoichiometric amounts of oxalyl dihydrazole and aluminum nitrate (both in DW) were then added to the gel. The product obtained was heated and maintained at around 300 °C. Finally, the material was ground and calcined at 500 °C for 2 h. The RGO-LaAlO_3_ composite was prepared in an autoclave reactor at 120 °C for 5 h.

The LaBO_3_ (B: Fe, Al, Mn, Co, and Ni) perovskites using a supercritical hydrothermal method were synthesized by Abe et al. [10], also who found the following optimal conditions to obtain a pure single phase of perovskite lanthanum aluminate: (i) La and Al nitrate solution at pH 8: (ii) Reaction temperature (450 °C), (iii) pressure (30 MPa), and (iv) time (15 min). To prepare their perovskites, these authors used the batch-type reactor vessel shown schematically in Figure 24a, and the process for preparing the solution mixture for this synthesis is shown in Figure 24b.

The mechanism of water and gas release during heat treatment of Eu^3+^-doped perovskite lanthanum aluminate using a kinetic thermodesorption mass spectrometry method was studied by Kreisberg et al. [82]. The main degassing products were H_2_O and CO_2_. The majority of water from the materials was released between 400 and 600 °C (with a maximum at 400 °C), whereas CO_2_ emission had a maximum at 600 °C and was mostly completed by 800 °C. 

The LaAlO_3_:Eu^3+^ perovskites prepared exhibited a size in the range 0.1–0.7 μm and the final product was pure LaAlO_3_. To prepare these materials, the authors used the thermovaporous method in a two-stage regime, namely in sub- and supercritical water. Thus, a stoichiometric mixture of Al(OH)_3_ and La_2_O_3_ along with the required amount of Eu_2_O_3_ and sufficient DW were poured into an autoclave reactor at 250 °C for 22 h. Then, the autoclave was then heated at 400 °C for 48 h.

Lee et al. [69] synthesized Eu^3+^-doped LaAlO_3_ perovskites at different concentrations (0, 1, 3, 5, 7, 9, and 11 mol%) using solvothermal synthesis. The as-prepared nanophosphors formed nanoparticles with high crystallinity. The optimum doping concentration of Eu^3+^ ions in LaAlO_3_ was about 9 mol%, and the optoelectronic characteristics shown by this material suggest that it could be useful for achieving natural white-light emission for application in WLEDs and FEDs. 

To prepare these materials, the authors dissolved stoichiometric amounts of the metal nitrates in 4 mL of deionized water then stirred the solution magnetically for 10 min. The mixtures prepared with various concentrations of Eu^3+^ and 4 mL of HNO_3_ solution (5 M) were mixed with 100 mL of ethylene glycol and stirred for 1 h. The resulting solutions were poured into an autoclave reactor and heated at 180 °C and maintained at that temperature for 10 h, after which time it was allowed to cool to RT. The precursors collected were washed with DI and dried. Finally, all samples were calcined at 800 °C for 2 h.

Hydrothermal and solvothermal routes have been used to produce ideal materials with controllable morphology and particle size as well as pure and small crystalline phases. Furthermore, hydrothermal synthesis in supercritical water is advantageous for scale-up preparation due to its short reaction time and the ability to use a continuous reaction apparatus. However, these methods require high temperature and pressure inside the autoclave.

## 4. Catalytic Performances of LaAlO_3_ Perovskites

### 4.1. Dry and Steam Reforming of Methane and Steam Reforming of Toluene, Glycerol, and Ethanol

Energy is a key element of our daily lives, and its demand is anticipated to rise steadily over time due to increasing world population and industrialization. In the past decade, the excessive use of fossil fuels has resulted in various environmental issues, such as a rapid rise in greenhouse gas emissions [102,103]. Consequently, the excessive amount of waste CO_2_ in the atmosphere has become a serious threat to our security and prosperity. The other gas contributing to the aforementioned problem is CH_4_, which has a 28-times stronger global warming potential (GWP) than CO_2_ [103]. 

In light of the above, technical approaches, such as catalytic conversion, sequestration, and capture have been applied to mitigate the emission of these two major greenhouse gases. Indeed, the DRM is currently a topic of great interest due to its remarkable ability to reduce greenhouse gases and convert them into a mixture of syngas (CO + H_2_), which can be subsequently converted into fuels, oxygenates, and hydrocarbons via the Fischer–Tropsch process [103]. 

Recent studies have showed that perovskites may serve as novel catalysts for syngas production via CO_2_ reforming of methane due to their intrinsic physicochemical properties, such as good storage and release of oxygen due to the formation of vacancies (oxygen and A or B cations), high thermal stability, properties, such as acidity and basicity, reducibility and strong metal–support interaction, which allows high resistance to sintering and carbon deposition [18,65,104]. 

Several supports, such as Al_2_O_3_ and La_2_O_3_, have been evaluated for DRM [18] as CO_2_ activation depends on the acid-base properties of the support, producing formates on an acidic support (Al_2_O_3_) and oxy-carbonates on an basic support (La_2_O_3_) [18]. In addition, the properties of aluminum oxides are improved by incorporating metals to their structure to obtain perovskites [19]. For example, the addition of La stabilizes the structure and changes the acid–base characteristics of γ-Al_2_O_3_ due to the formation of LaAlO_3_ perovskite.

Another topic of interest worldwide is related to the generation of hydrogen, which is well-known to have a higher energy content than any common fuel and to offer a clean and environmentally friendly solution to solve energy requirements in the coming years [104]. In this sense, LaAlO_3_ perovskites have recently also been used as catalysts in the steam reforming of methane (SRM), toluene, glycerol, and ethanol to generate H_2_. Each of the aforementioned catalytic applications are discussed below.

#### 4.1.1. Dry Reforming of Methane to Syngas

The partial substitutions of A or B cations with specific metals allow to form oxygen vacancies in perovskite, which plays a key role in the catalytic performance and resistance of catalysts to carbon deposition. Thus, some noble metal (Rh, Ru, Ir, Pt, and Pd)-incorporated LaAlO_3_ perovskites have been reported to be active for the DRM. For example, Anil et al. [18] synthesized LaAlO_3_ and Ru-, Pt-, and Pd-substituted LaAlO_3_ and evaluated the catalytic performance of these catalysts for the DRM. They found that LaAl_0.98_Ru_0.02_O_3−δ_ showed the highest conversion for both methane (100%) and carbon dioxide (86%) at 800 °C with an H_2_/CO molar ratio close to 0.85. 

Furthermore, this catalyst showed a stable conversion for 50 h. A set of perovskite catalysts LaAl_1−x_Ni_x_O_3_ (0 ≤ x ≤ 1) and their counterparts modified by partial substitutions with noble metals (La_0.4_Al_0.2_Ni_0.8_M_0.6_O_3_, M = Pt, Pd, Ru, Rh, and Ir) were synthesized by Arandiyan et al. [105]. According to these authors, the CH_4_ and CO_2_ conversions remained close to 90% and 40%, respectively, and the H_2_/CO molar ratio remained close to 1. The use of La_0.4_Al_0.2_Ni_0.8_M_0.6_O_3_ as starting material led to highly active and stable catalysts for the DRM. 

However, due to the high cost of preparing catalysts modified with noble metals, combined with the scarcity of the latter, their application on an industrial scale has been limited. As such, researchers are currently looking for alternative metals for use in catalysts that are abundant in nature and cheap to extract for use on an industrial scale. Alkaline earth metals, such as Mg, and transition metals, such as Ni, have been shown to be active for the DRM. Recently Bai et al. [65] prepared LaAlO_3_ and La_1−x_Mg_x_AlO_3−δ_ and used these materials to incorporate active sites of nickel for DRM. They found that an Mg/La molar ratio of 2/8 was the optimal content of Mg substitution (Ni/La_0.8_Mg_0.2_AlO_2.9_). CH_4_ conversion using this catalyst reached 75.4%. 

The results were explained by the low sintering Ni nanoparticles and because the partial substitution of Mg induced more basic sites and more oxygen vacancies, which facilitated carbon gasification and thus suppressed carbon deposition. Sagar et al. [79] prepared two series of LaNi_x_Al_1−x_O_3_ perovskites (0 ≤ x ≤ 1) using hydrothermal and sol–gel methods. According to the catalytic DRM results, the catalytic performance for the best catalyst prepared by the hydrothermal method (LaNi_0.6_Al_0.4_O_3_) was better than that of the best catalyst prepared by the sol–gel method (LaNi_0.3_Al_0.7_O_3_). Thus, the hydrothermal catalyst gave CH_4_ and CO_2_ conversions of 94% and 97%, respectively, with an H_2_/CO ratio of 0.98, while the sol–gel catalyst gave CH_4_ and CO_2_ conversions of 93% and 96%, respectively, and an H_2_/CO ratio of 0.97.

Figueredo et al. [19] also synthesized and evaluated LaAlO_3_ as a catalytic support in DRM using Ni as an active phase and compared their results with those active sites of nickel incorporated on commercial α-Al_2_O_3_. The results of the catalytic tests indicated that the Ni/LaAlO_3_ catalyst was 7.8% and 11.5% more stable than Ni catalyst supported on commercial α-Al_2_O_3_ for the conversion of methane and carbon dioxide, respectively. 

These authors also found that the presence oxygen vacancies and reducibility of the NiO NPs increased its catalytic performance and stability and that the presence of carbon nanotubes increased the metallic dispersion (see Figure 25). Furthermore, in previous studies, Urasaki et al. [106] determined that Ni supported on LaAlO_3_ perovskite showed a much higher coking resistance than did Ni/α-Al_2_O_3_. This is due to the fact that the lattice oxygen on the LaAlO_3_ perovskite can oxidize the carbon (C) and CH_x_ fragments adsorbed on metal active sites, thereby, accelerating the rate of the chemical reaction and suppressing carbon deposition.

#### 4.1.2. Steam Reforming of Methane to Generate H_2_

Ou et al. [104] evaluated the catalytic performance of Ni catalysts in SRM as a function of H_2_ production, CH_4_ conversion, and catalyst stability. Thus, these authors prepared several La_0.7_A_0.3_AlO_3−δ_ (A = Ca, Ba, Ce, Zn, Sr, and Mg), LaBO_3_ (B = Al, Fe, and Mn), and La_1−x_Mg_x_AlO_3−δ_ catalysts. Their results showed that modification with Ba, Ca, Ce, or Zn gave lowers the catalytic performance, whereas the partial incorporation of Sr or Mg improved the catalytic performance of Ni-supported lanthanum aluminate catalyst (the CH_4_ conversion increased from 67.5% to 69.5% and 72.1%, and the H_2_ yield from 51.3% to 56.6% and 61.2%, respectively; see Figure 26). 

More importantly, these authors showed that the Ni/La_0.7_Mg_0.3_AlO_3−δ_ catalyst exhibited the highest catalytic performance (CH_4_ conversion of 72.1% and H_2_ yield of 61.2%) and high suppression to carbon deposition, remaining stable after 35 h. Yang et al. [107] elucidated the effect of the perovskite-type structure, the partial substitution in B-site (LaBO_3_, B = Al, Cr, and Fe), and the active sites-support interaction on the SRM (steam reforming of methane) catalytic activity. To avoid nickel sintering, the Ni added was less than 2 wt.%. 

These authors found that the catalytic performance of the catalysts was mainly due to the active site surface area in the following order: 2%Ni/LaAlO_3_ > 2%Ni/LaCrO_3_ > 2%Ni/LaFeO_3_. However, the ability to suppress carbon deposition was mainly governed by the amount of oxygen vacancies according to the following order: 2%Ni/LaCrO_3_ > 2%Ni/LaAlO_3_ > 2%Ni/LaFeO_3_. Out of the catalysts prepared, 2%Ni/LaAlO_3_ possessed the highest nickel surface area, thus, showing the highest catalytic activity at 800 °C (CH_4_ conversion and H_2_ yield of approximately 70% and 40%, respectively).

Mukai et al. [108] determined the catalytic activity of a Pt catalyst supported on La_1−x_Sr_x_AlO_3−0.5x_ for the partial oxidation of CH_4_ and compared their results with those obtained for Pt/α-Al_2_O_3_. Pt/La_0.7_Sr_0.3_AlO_2.85_ was found to provide a higher and more stable hydrogen yield after a reaction for 190 min. The hydrogen yield remained stable at around 40% due to the high Pt dispersion.

#### 4.1.3. Steam Reforming of Toluene to Generate H_2_

The catalytic activity of several catalysts synthesized by modification of the Ni/LaAlO_3_ perovskite with various alkaline-earth metals (Ca, Sr, and Ba) in the steam reforming of toluene was evaluated by Higo et al. [109]. The addition of Ba (Ba/Ni/LaAlO_3_) was found to markedly promote the catalytic activity and tolerance against oxidation. This catalyst was able to convert toluene with an H_2_ yield of more than 40%. Oemar et al. [110] studied the catalytic performance in the steam reforming of tar with toluene to obtain syngas using Ni/LaAlO_3_, Ni/La_0.8_Sr_0.2_AlO_3_, Ni/La_2_O_3_, and Ni/α-Al_2_O_3_ as catalysts. 

Among the materials studied, Ni/La_0.8_Sr_0.2_AlO_3_ showed the highest catalytic activity as function of both the catalytic performance (average toluene conversion and H_2_ yield of 20.0% and 66.5%, respectively) and coke resistance (12.8% without O_2_ and 5.0% with O_2_) at 650 °C. In addition, these authors demonstrated that the better catalytic performance of this catalyst was due to the fact that the partial substitution of Sr on crystal structure induced a high amount of oxygen vacancies.

Mukai et al. [111] investigated the catalytic performance in the steam reforming of toluene to obtain H_2_ using Ni/LaAlO_3_, Ni/La_0.7_Sr_0.3_AlO_3−δ_, and Ni/α-Al_2_O_3_ as catalysts. Among the catalysts evaluated, Ni/La_0.7_Sr_0.3_AlO_3−δ_ exhibited the highest activity (toluene conversion of 58.2% and hydrogen yield of 48.4%) and lowest coke formation (57 mg/g_cat_). This was due to the fact that the surface lattice oxygen in the catalyst participated in the redox mechanism. Furthermore, the carbon deposited on the catalyst surface after the catalytic reaction was removed through oxidation reaction by lattice oxygen in/on the support. 

These authors demonstrated that the partial substitution of Sr on crystal structure induced a high amount of oxygen vacancies, which enhanced its redox ability. In another study, Mukai et al. [112] also evaluated the role of the interaction between Ni and the lattice oxygen in Ni/La_0.7_Sr_0.3_AlO_3−δ_ catalysts sintered at different temperatures (800, 900, 1000, and 1100 °C), for H_2_ production by steam reforming of CH_4_. These authors determined that sintering of the Ni particles increased (from 13.9 to 182.6 nm) with calcination temperature (see Figure 27), and the C_1_ yield decreased from 72.9% to 15.5%. 

In addition, Mukai et al. [113] studied the effect of Pt incorporation to Ni/La_0.7_Sr_0.3_AlO_3−δ_ functional material on the catalytic toluene steam reforming for H_2_ production. Their results revealed that Pt addition enhanced the catalytic activity (carbon yield of 59.1% and hydrogen yield of 52.7%, without pre-reduction at 10 min of reaction) and showed a high resistance to coke formation (8 mg/g_cat_ after 10 min of reaction).

#### 4.1.4. Steam or Aqueous Phase Reforming of Glycerol to Generate H_2_

Kim et al. [114] studied H_2_ production from glycerol by SR using Ni-Ce-supported LaAlO_3_ perovskite and evaluated the effect of the of Ce loading amount. Among the catalysts evaluated, the 15Ni-5Ce/LaAlO_3_ catalyst (15 wt% Ni and 5 wt% Ce) showed the highest glycerol conversion (94.6%) and H_2_, CO_2_, and CH_4_ selectivities of 63.4%, 96.6%, and 2.9%, respectively. The high performance activity of this catalyst was attributed to the amount of cerium added to the support. 

Moreover, the smaller particle size and the high number of active sites played a key role in the catalytic performance and hydrogen selectivity. Park et al. [115] also investigated the catalytic activity of Ni-supported LaAlO_3_ perovskite (X–Ni/LaAlO_3_ (X = Cu, Co, and Fe); (X = 5% wt.%)) for the glycerol aqueous-phase reforming to produce hydrogen. Among the catalysts evaluated, the Cu–Ni/LaAlO_3_ catalyst exhibited the highest glycerol conversion (34.6%) and H_2_ selectivity (79.7%), which was due to the synergistic effect of the Ni and Cu metals in the functional material. 

Lee et al. [116] examined the aqueous phase reforming of glycerol using four Ni-supported perovskites (Ni/LaAlO_3_, Ni/CeO_2_, Ni/MgO, and Ni/MgAl) and found that the Ni/LaAlO_3_ catalyst showed the highest catalytic activity (glycerol conversion of 36%, H_2_ and CO_2_ selectivity of 96% and 81%, respectively) and also displayed good stability. These authors also showed that the high resistance of the Ni/LaAlO_3_ catalyst to coking was related to the migration of mobile reactive oxygen from the crystal lattice of LaAlO_3_ perovskite to the metallic Ni nanoparticles.

Kim et al. [117] investigated the hydrogen production process in the aqueous phase reforming of the glycerol by using Ni-Cu-supported LaAlO_3_ catalysts in order to evaluate the effects of different Cu loading (0, 3, 5, 7, and 10 wt%). They showed that the 15Ni-5Cu/LaAlO_3_ catalyst gave the highest glycerol conversion (34.6%) and hydrogen selectivity (79.7%) as it contained the optimal copper loading.

#### 4.1.5. Ethanol Steam Reforming to Generate H_2_

Recently, Martinez et al. [66] explained the role of the LaAlO_3_-based support on the Rh nanoparticles properties in the ethanol SR reaction. In addition, they found that the partial substitution of La with Ce and Ca in the support structure modified its porosity and surface features. Thus, the inclusion of Ca in the structure increased the S_BET_ of the support (from 10 m^2^/g to 22 m^2^/g), whereas Ce inclusion activated the mobile reactive oxygen of crystal lattice of the catalyst. 

In addition, the supports influenced the degree of reduction of rhodium nanoparticles incorporated at the surface of the catalysts. Among the catalysts evaluated, the 0.3%Rh/LaAlO_3_ catalyst exhibited the highest mean value of H_2_ product distribution (80%) a low amount of byproducts (CH_4_ and CO, less than 10% in both cases) with outstanding stability over a period of 24 h of chemical reaction.

Ohno et al. [118] prepared Co-doped LaAlO_3_ nanoparticles with partial reduction and used them as catalysts in ethanol SR. The La(Al_0.5_,Co_0.5_)O_3_ catalyst prepared using a chemical solution deposition method showed the highest H_2_ yield (78%).

### 4.2. Oxidative Coupling of Methane

Methane, which is an abundant hydrocarbon and the principal component of natural gas has aroused worldwide interest as a primary source of fuels and chemicals [119,120,121]. This and the increased exploiting of natural gas from unconventional reservoirs have led to renewed interest in converting methane into value-added chemicals. This explains the importance of OCM, which consists of the direct catalytic conversion of methane into higher hydrocarbons, such as ethylene—that is, the direct catalytic reaction between methane and oxygen over catalysts to give C_2_ products (ethylene (C_2_H_4_) and ethane (C_2_H_6_)) [120].

Sim et al. [63] evaluated the catalytic behavior of LaAlO_3_ catalysts in OCM, and they found that the best prepared catalyst (LaAlO_3__8; co-precipitation method with pH = 8) showed a high C_2_ yield (16%) due the electrophilic lattice oxygen and the presence of oxygen vacancies. These authors also found an optimum reaction temperature range (between 750 and 800 °C) to maximize the OCM reaction using lanthanum aluminate perovskite catalysts.

Lee et al. [69] prepared LaAlO_3_ catalysts with different crystallinities using three preparation methods (citrate sol–gel (C), solid-state (S), and hard templating method (H)) and studied the relationship between the crystallinities and catalytic activity for OCM. Among the catalysts evaluated, the catalyst prepared with the C method showed the best catalytic performance (methane conversion and C_2_ selectivity of 24% and 50%, respectively), which was due to the 100% relative crystallinity of this catalyst favoring a high retention of electrophilic lattice oxygen species.

Kim et al. [122] also investigated the effect selective lattice oxygen species of three kinds of lanthanum-based perovskite in the catalytic performance of OCM at different temperatures (500–800 °C) either with or without gaseous oxygen. Among the catalysts evaluated, they found that the LaAlO_3_ catalyst showed the highest C_2_ yield (10.0%) at 750 °C as it contained abundant electrophilic surface lattice oxygen species due to the facile filling of lattice oxygen vacancies by gaseous oxygen.

Other researchers have evaluated the catalytic behavior of various catalysts prepared by doping LaAlO_3_ perovskite with several alkali metals due to their basic properties in the OCM reaction. For example, An et al. [73] added alkali metals (Li, Na, and K) to the LaAlO_3_ perovskite and demonstrated that the catalytic performance for OCM improved with the load of the incorporated alkali metal. They showed that the resulting catalysts by adding alkali metals exhibited a 2–8% higher activity than the LaAlO_3_ catalyst. In addition, the authors showed that catalysts with high C_2_H_4_ selectivity contained high retention of electrophilic lattice oxygen species, while the catalysts with high CO_x_ selectivity contained high retention nucleophilic oxygen species.

Kwon et al. [123] used a large amount of lithium as an additive during the preparation of a lanthanum aluminate catalyst to form another phase, namely La_2_Li_0.5_Al_0.5_O_4_ (K_2_NiF_4_-type structure), which was found to exhibit outstanding catalytic performance in the OCM reaction. These authors evaluated the catalytic reaction at various times (1, 10, and 80 h; see Figure 28) and determined that, after 80 h of OCM, La_2_Li_0.5_Al_0.5_O_4_ gave a higher C_2_ yield (15%) compared with LaAlO_3_ (12%). The high C_2_-selectivity obtained with La_2_Li_0.5_Al_0.5_O_4_ was due to its high content electrophilic surface lattice oxygen species.

In order to achieve methane conversions at lower temperatures, unconventional catalytic systems have been developed through the use of an electric field during catalytic reactions. In this sense, Sato et al. [67] studied the catalytic performance of OCM in an electric field of 3 mA at 150 °C, over M-substituted LaAlO_3_ catalysts (M = Ba, Ca, and Sr) and found that La_0.7_Ca_0.3_AlO_3−δ_ gave the highest C_2_H_6_ + C_2_H_4_ yield (11.1%) and selectivity (20%). Moreover, this catalyst was stable for 180 min, with no structural deformations or carbon deposition. In this same order of ideas, Yabe et al. [68] investigated the catalytic performance of OCM in an electric field of 3 mA at different temperatures (from 150 °C to 360 °C) over Ca-doped LaAlO_3_ catalysts. They found that La_0.7_Ca_0.3_AlO_3−δ_ exhibited the highest C_2_ yield (7.4%).

### 4.3. Three-Way Catalysts

The emission of exhaust gases from vehicles has become one of the main primary air pollutants. Indeed, given the increasing numbers of vehicles being used around the world, the problems of these emissions are becoming more complex. As such, the control and treatment of pollution derived from vehicle exhaust gases has become a highly relevant topic [124]. TWC have long been recognized as an economic and efficient solution for removing different air pollutants, such as NO_x_, hydrocarbons (HC), and carbon monoxide (CO) from gasoline engine exhaust gases under near-stoichiometric conditions. 

Furthermore, perovskite-type solids have been studied as a promissory TWC material, principally to enhance its thermal stability [72]. Among various candidates for TWC supports, LaAlO_3_ perovskite has been considered a potential material. Thus, Tran et al. [20] studied different Co-doped LaAlO_3_ perovskites for NO_x_-assisted soot oxidation. The results revealed that Co was incorporated into the perovskite lattice. They also found that adding Co up to 75% significantly improved the oxidation reaction, whereas replacing 100% of Al by Co deteriorated the catalytic activity. 

Among the catalysts prepared, LaAl_0.25_Co_0.75_O_3_ was found to be the most active for NO oxidation at 320 °C (conversion of 78%) and also showed the highest activity for soot oxidation in the presence of NO_x_ gases (with T_10%_ of 377 °C), maintaining high nitrogen dioxide production (71%) (see Figure 29). This catalytic behavior of the best catalyst was associated with high-migration reactive oxygen species.

Higo et al. [125] investigated NO reduction at low temperature by propylene using a Pd/La_0.9_Ba_0.1_AlO_3−δ_ catalyst. This catalyst showed the highest catalytic performance (NO conversion of 46.5%) at low temperatures (300 °C or lower). In this case, the electrophilic surface lattice oxygen species contributed to obtain of intermediates compounds (oxygenated species (C_x_H_y_O_z_)) and accelerated the reduction of nitrogen monoxide. Yoon et al. [72] studied the TWC catalytic performance and thermal stability of Pd-incorporated LaAlO_3_ perovskite catalysts, and these were compared with Pd-incorporated γ-Al_2_O_3_ catalysts. 

The results obtained showed that the perovskite catalysts exhibited higher TWC catalytic performance and better thermal stability than γ-Al_2_O_3_ catalysts under reaction condition similar to gasoline engine exhaust. This was attributed to the promoting effect of lanthanum in the perovskite crystal lattice for TWC reactions. In addition, Pd sintering was suppressed during the thermal treatment due to the strong La-Pd interaction induced by the effective electron transfer of lanthanum to palladium.

## 5. Summary and Future Perspectives

Several methods for preparing LaAlO_3_ perovskites have been described in the current work review. Although solid-state synthesis is a conventional and simple method for synthesizing these materials, this procedure still has several serious drawbacks—for example, the need for a high reaction temperature, the introduction of impurities during milling, secondary-phase formation, large particle size, and broad particle distribution. 

Extensive scientific studies have been conducted to optimize the production of such perovskites using diverse synthetic chemical methods. Synthetic processes at lower temperatures are necessary as they lead to small particle sizes and high specific surface areas. Recently, a phase-pure LaAlO_3_ perovskite with an average particle size of about 29 nm was synthesized by way of a sol–gel reaction using natural reagents in which the calcination temperature was reduced to 600 °C [8]. 

It is apparent from this review that chemical co-precipitation using aqueous Na_2_CO_3_ solution as an agent precipitant is an easy route for the preparation of LaAlO_3_ perovskites at pH values of 7 and 8 [63]. The lowest calcination temperature used to date for the synthesis of LaAlO_3_ perovskite was 400 °C using the thermovaporous method in water, although with a large particle size (100–700 nm) [82]. The nanoparticle sizes reported to date range from 11 to 700 nm with calcination temperatures ranging from 400 to 1700 °C. The temperature and time reactions for the synthesis of LaAlO_3_ perovskites have been reduced using LaCO_3_OH instead of La_2_O_3_ and by adding metal fluorides to the solid-state preparation. 

Furthermore, the molten salt synthesis method and shock-assisted solid-state method have proven advantageous for the preparation of such materials as they meet important requirements, such as a short preparation time and being simple, facile, high-yielding, cost-effective, and environmentally friendly. However, the citrate sol–gel synthesis is still the method of choice for researchers to prepare certain members of the LaAlO_3_ perovskite family due to its advantages, namely high homogeneity and purity as well as good control of the composition of the final material.

Irrespective of the method chosen, the XRD results showed that no intermediate La(OH)_3_, La_2_O_3_, Al(OH)_3_, or Al_2_O_3_ phase was present. In most cases, pure LaAlO_3_ adopted a rhombohedral crystal structure with the space group R-3c also showing particles with several morphologies (spherical, semi-spherical, irregular spherical, polygonal, hexagonal, grain compacted, small blocks, spongy, and perforated cage).

LaAlO_3_ perovskite exhibits excellent thermal and chemical stability, as well as interesting optoelectronic properties, thus, leading to a wide variety of possible applications. For example, this material has recently been evaluated as a catalyst, for microwave absorption applications, as an adsorbent to remove MO, and as a nano-adsorbent for the removal of fluoride and dyes (such as the DB-53 dye), and its antimicrobial activity against microorganisms has also been studied in vitro. 

Furthermore, given that LaAlO_3_ offers wide versatility for the substitution of La^3+^ and Al^3+^, it has recently been used as a catalyst in various types of reactions (DRM, SRM, steam reforming of toluene, glycerol and ethanol, OCM, and TWC), in medical applications (such as the targeted drug delivery and bioimaging of breast cancer cells), for the oxidative degradation of organic pollutants, as an adsorbent, for TL dosimetry applications, and for water-based acrylic paints.

Although the preparation methods published to date have proven successful for obtaining some representative members of the LaAlO_3_ perovskite family, they are not completely satisfactory since these present some disadvantages, such as the use of environmentally unfriendly and costly precursors and solvents. 

Therefore, and given the need to synthesize pure and homogeneous perovskite-type materials, novel and better methods are still required to reduce the calcination temperatures that allows the suppression of sintering, thus, increasing the number of metals that can be stably incorporated and increasing their dispersion, thereby, making the active centers more accessible for practical applications, such as DRM, SRM, steam reforming of toluene, glycerol and ethanol, OCM, and TWC. These two objectives will allow future lines of research to focus on optimizing the production routes to this kind of perovskite.

In conclusion, there are still several interesting fields to be explored regarding the preparation of representative members of the LaAlO_3_ perovskite family that allow such materials to be obtained in a simple, rapid, high-yielding, facile, cost-effective, and environmentally friendly manner. For example, the synthesis of these materials using industrial metal wastes as precursors may be of relevant interest to implement green chemistry principles in the circular economy.

## Figures and Tables

**Figure 1 materials-15-03288-f001:**
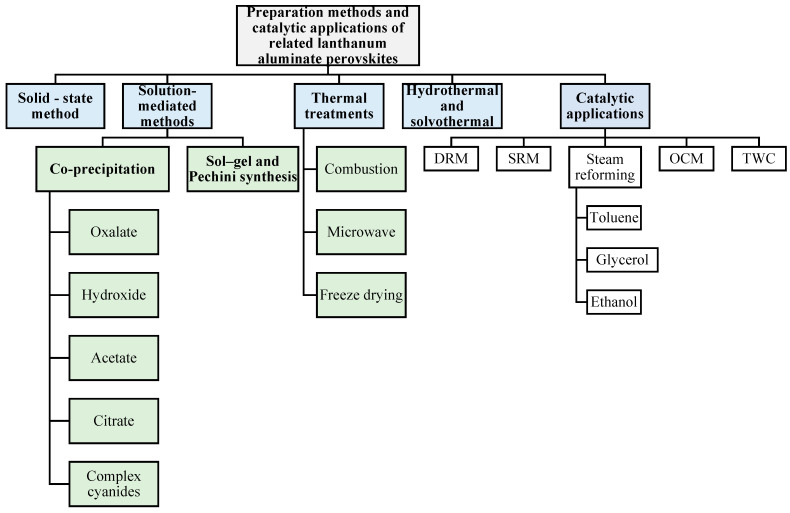
Synthesis and catalytic applications of related LaAlO_3_ perovskites.

**Figure 2 materials-15-03288-f002:**
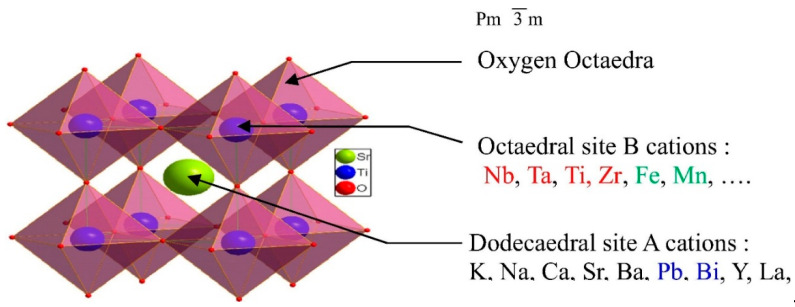
Cubic perovskite SrTiO_3_ (Reprinted with permission from [5]).

**Figure 3 materials-15-03288-f003:**
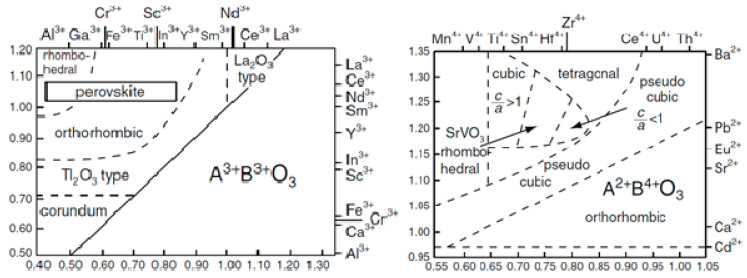
Effects of the ionic size of A and B cations on the distortions observed in the crystal structure of perovskite for A^2+^B^4+^O_3_ and A^3+^B^3+^O_3_ combinations (Adapted from [25]).

**Figure 4 materials-15-03288-f004:**
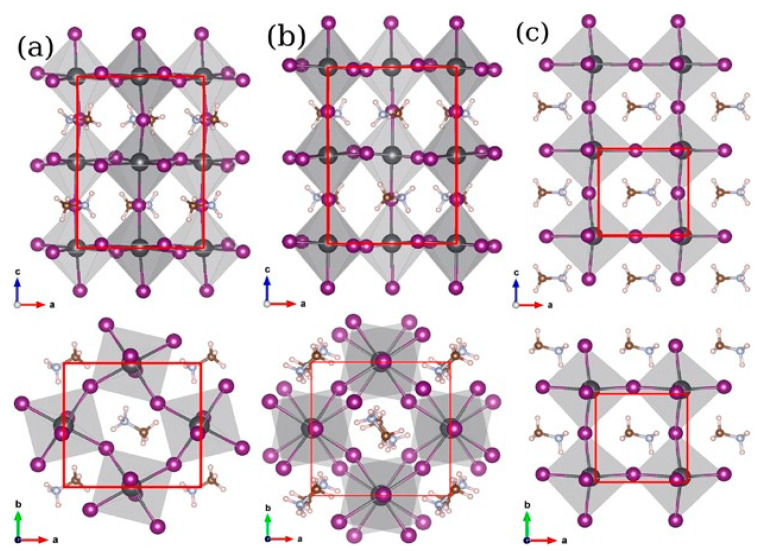
Different phases of perovskite (MAPbI_3_): (**a**) orthorhombic, (**b**) tetragonal, and (**c**) cubic. Top row: (a-c) plane and bottom row: (a-b) plane (Reproduced from [27]).

**Figure 5 materials-15-03288-f005:**
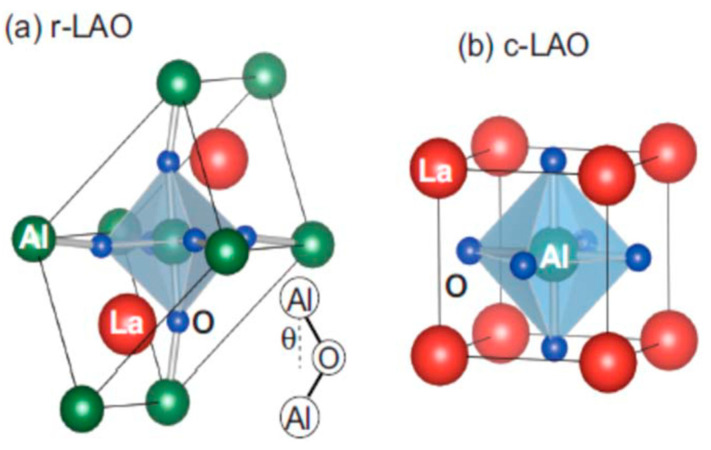
Crystal structures of LaAlO_3_ perovskite: (**a**) rhombohedral (r-LaAO) and (**b**) cubic (c-LaAlO) (Reprinted with permission from [29]).

**Figure 6 materials-15-03288-f006:**
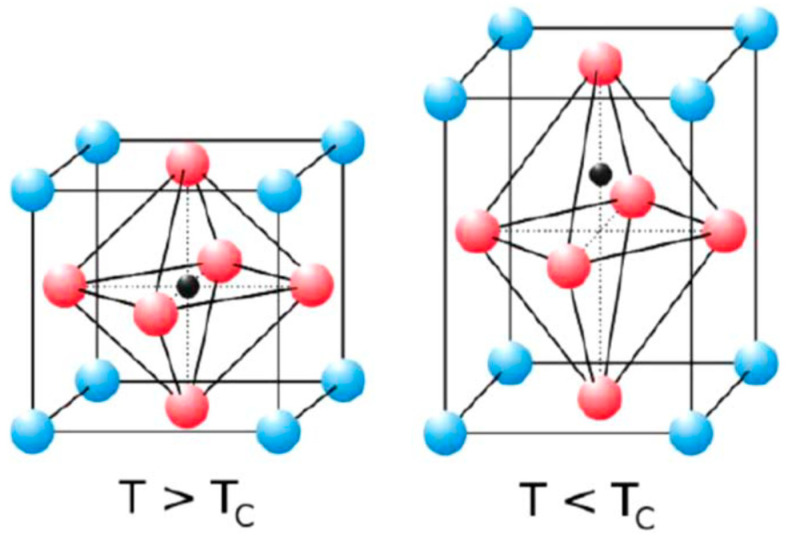
Structure of LaAlO_3_. La^3+^ is the cation placed in the corners (at site A, blue ball). Al^3+^ is the cation placed in the center (at site B, black ball) (Reprinted with permission from [29]).

**Figure 7 materials-15-03288-f007:**
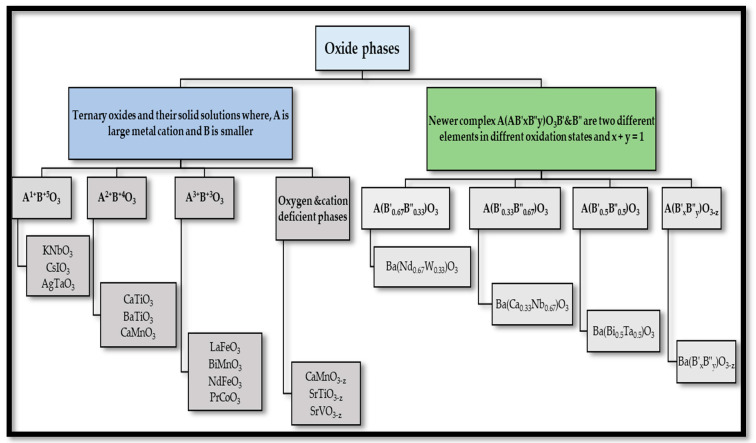
Classification of perovskite structures (Reprinted with permission from [5]).

**Figure 8 materials-15-03288-f008:**
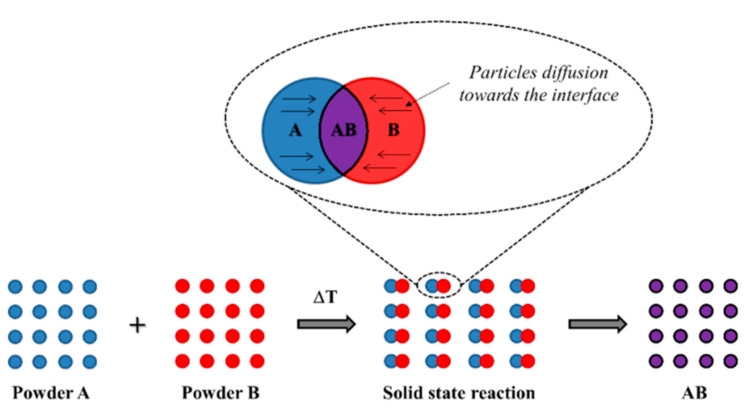
Schematic showing particle diffusion during the solid-state reaction (Reprinted with permission from [42]).

**Figure 9 materials-15-03288-f009:**
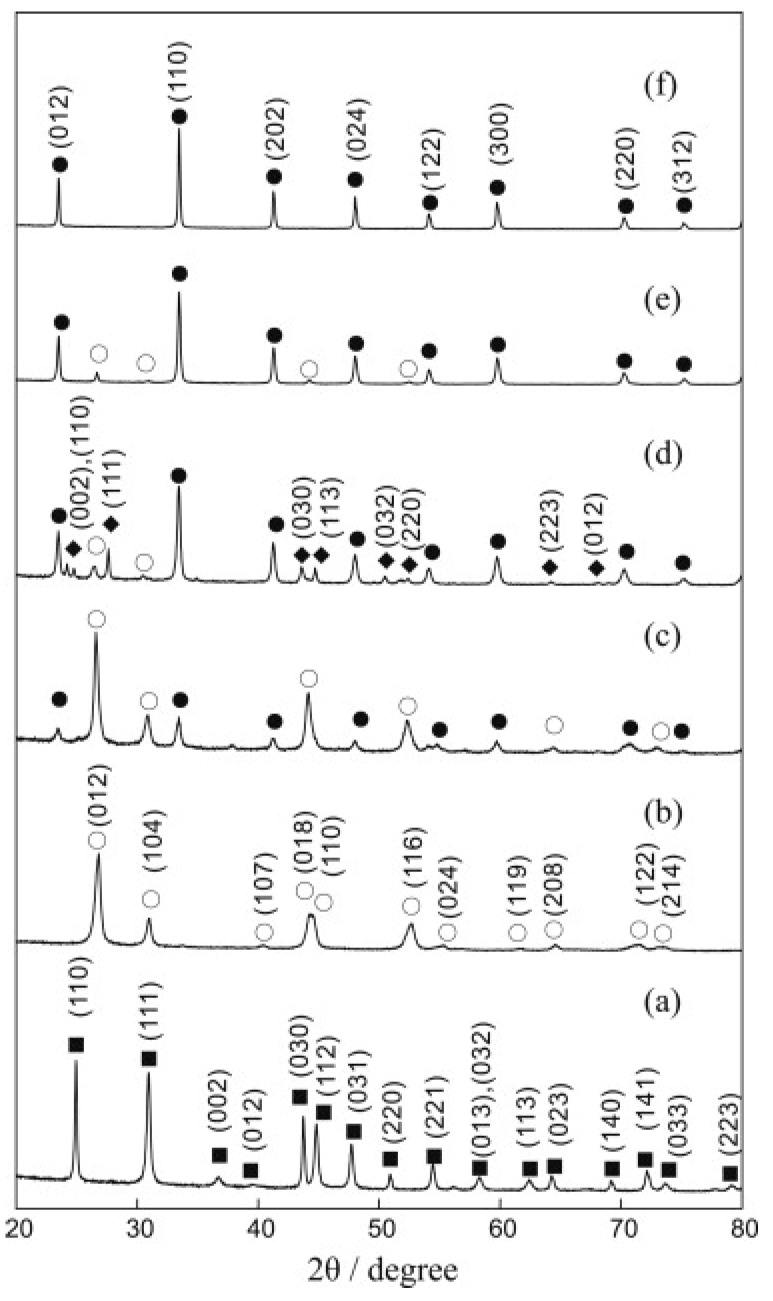
XRD patterns of powders obtained by the calcination of (**a**) a mixture of LaFCO_3_ and α-Al_2_O_3_ at (**b**) 800 °C, (**c**) 850 °C, (**d**) 900 °C, (**e**) 1000 °C, and (**f**) 1100 °C for 5 h under an air atmosphere. (■) LaFCO_3_, (○) r-LAOF, (◊) α-Al_2_O_3_, (♦) LaF_3_, and (●) h-LaAlO_3_ (Reprinted with permission from [44]).

**Figure 10 materials-15-03288-f010:**
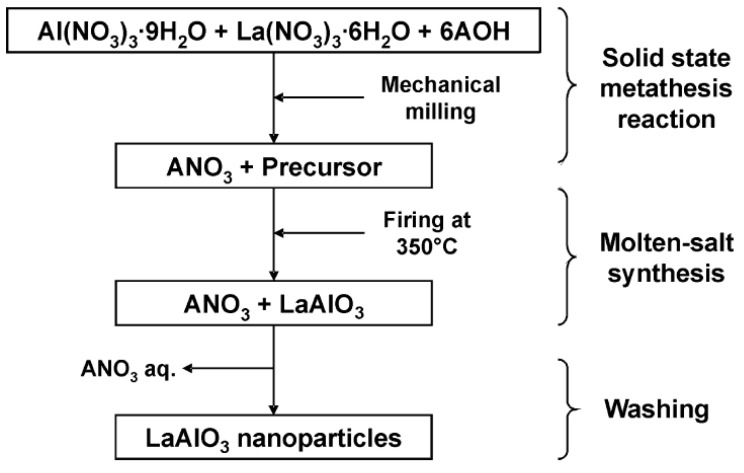
Flowchart of the experimental procedure for the synthesis of LaAlO_3_ (Reprinted with permission from [51]).

**Figure 11 materials-15-03288-f011:**
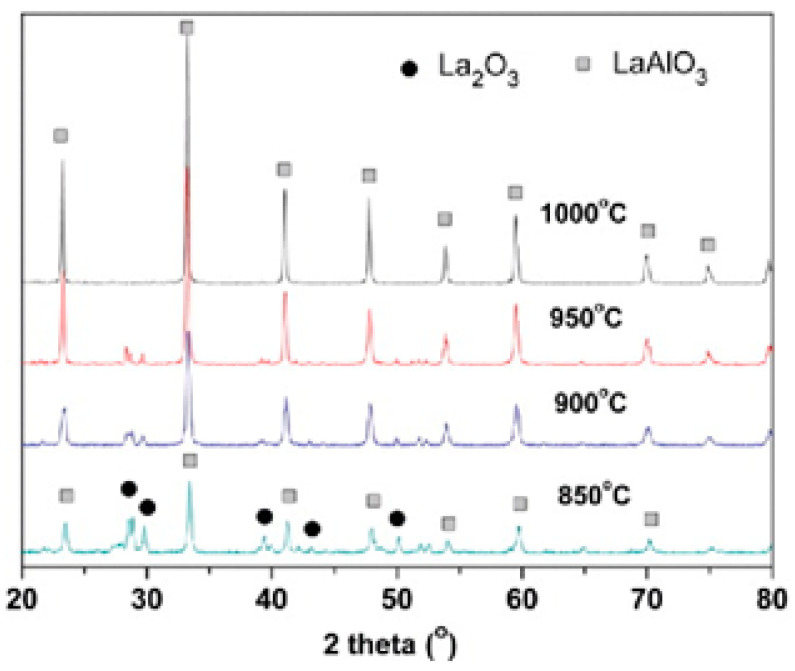
XRD patterns of LaAlO_3_ perovskites obtained by molten salt synthesis in the range of 850–1000 °C for 3 h. (Adapted from [54]).

**Figure 12 materials-15-03288-f012:**
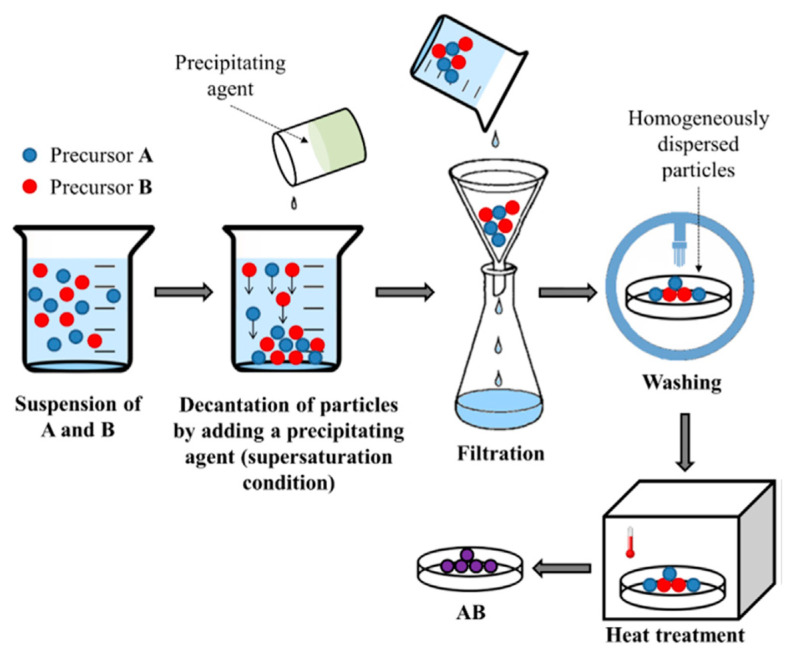
General schematic of the co-precipitation method (Reprinted with permission from [42]).

**Figure 13 materials-15-03288-f013:**
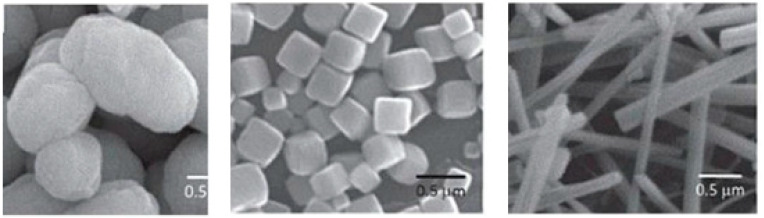
Co-precipitated perovskites with various crystal structures: (**left**) orthorhombic LaFeO_3_, (**center**) (Cubic SrTiO_3_), and (**right**) hexagonal LaAlO_3_, and (Adapted from [60]).

**Figure 14 materials-15-03288-f014:**
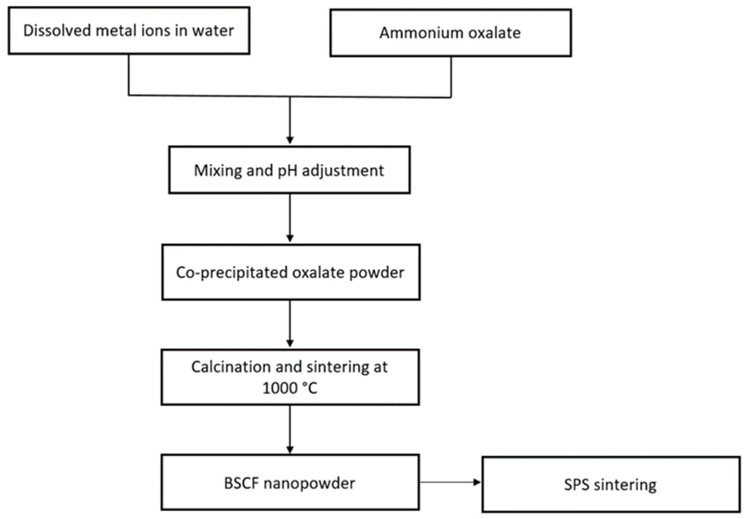
Schematic procedure for the synthesis of BSCF powder by the solution co-precipitation method (Reprinted with permission from [85]).

**Figure 15 materials-15-03288-f015:**
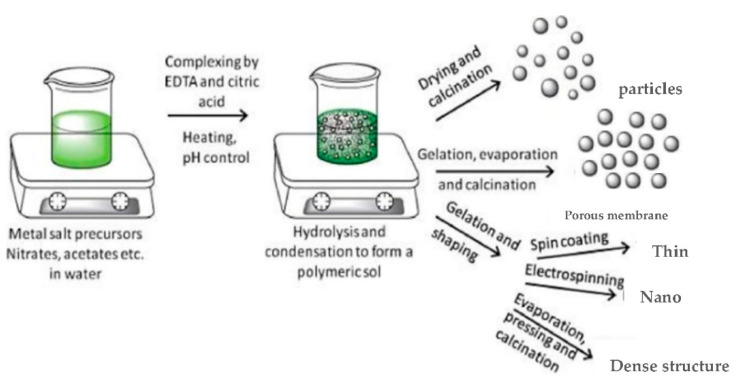
Steps in the sol–gel process to achieve final morphologies (Adapted from [60]).

**Figure 16 materials-15-03288-f016:**
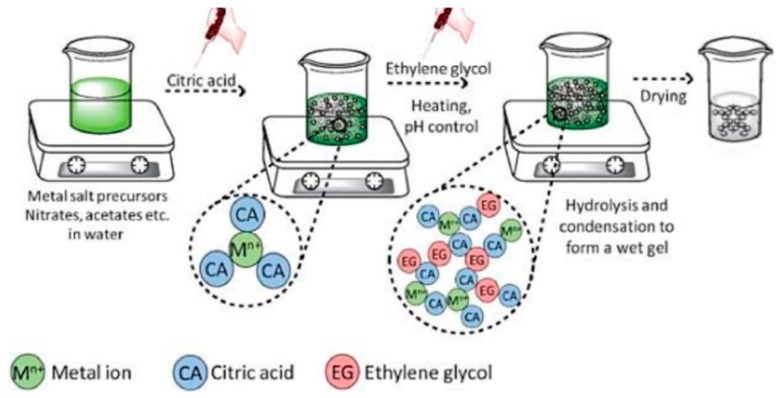
Schematic representation of Pechini’s method (Adapted from [60]).

**Figure 17 materials-15-03288-f017:**
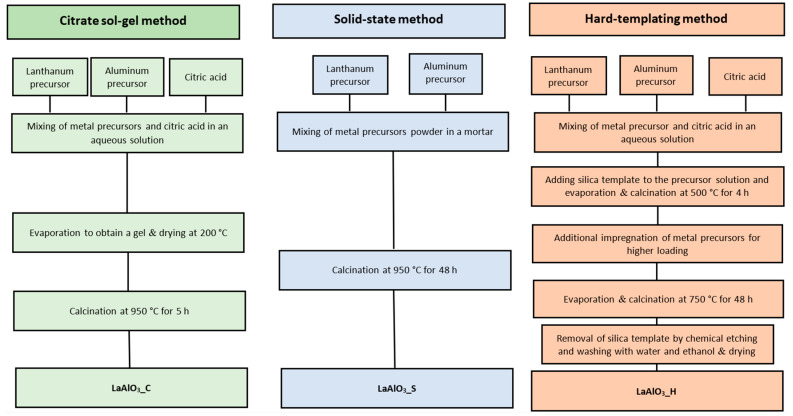
Schematic representation of the preparation methods for obtaining LaAlO_3_ perovskite catalysts (Reprinted with permission from [69]).

**Figure 18 materials-15-03288-f018:**
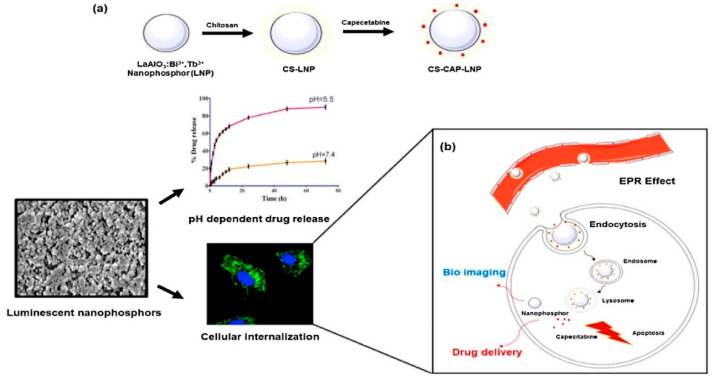
(**a**) Synthesis of CS-CAP-LNP. (**b**) Accumulation of CS-CAP-LNP at the tumor site due to the EPR effect and internalization of CS-CAP-LNP in tumor cells (Reprinted with permission from [70]).

**Figure 19 materials-15-03288-f019:**
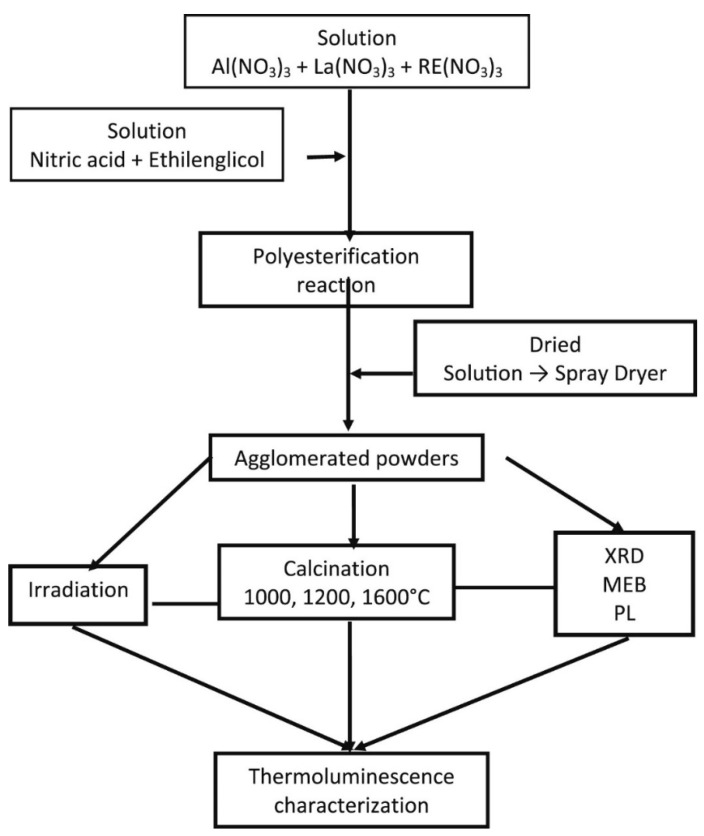
Flowchart for preparing LaAlO_3_:RE using the modified Pechini method (Reprinted with permission from [97]).

**Figure 20 materials-15-03288-f020:**
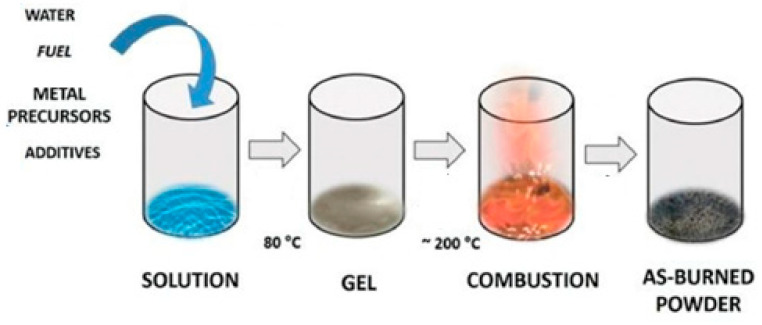
Schematic showing the steps in a typical combustion process (Adapted from [60]).

**Figure 21 materials-15-03288-f021:**
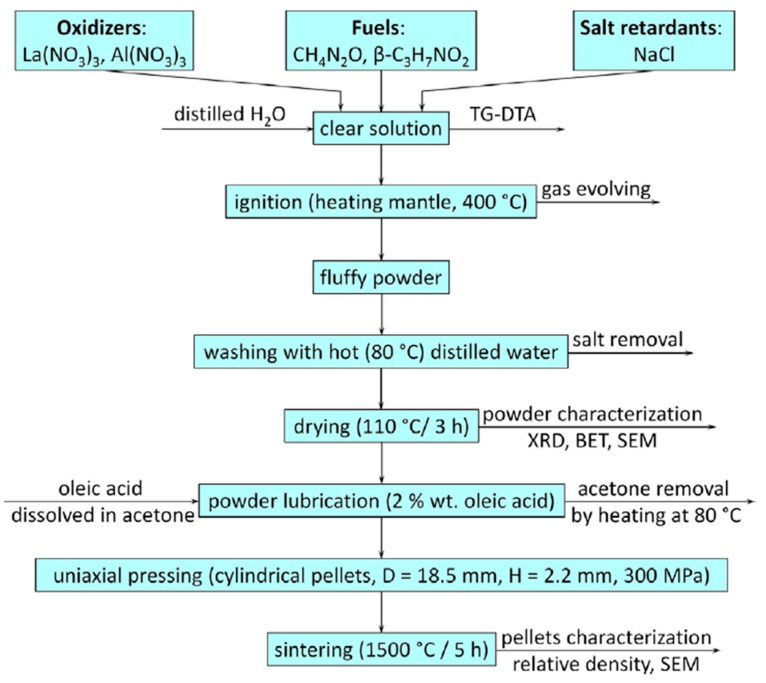
Flowchart showing LaAlO_3_ perovskite preparation, sintering, and characterization (Reprinted with permission from [77]).

**Figure 22 materials-15-03288-f022:**
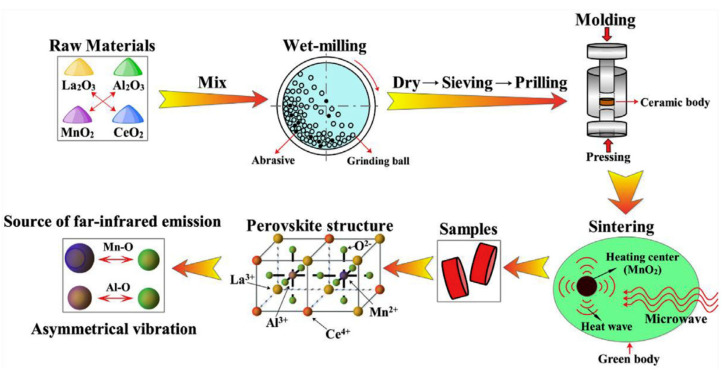
Schematic diagram for the synthesis of Ce/Mn-doped LaAlO_3_ perovskite (Reprinted with permission from [78]).

**Figure 23 materials-15-03288-f023:**
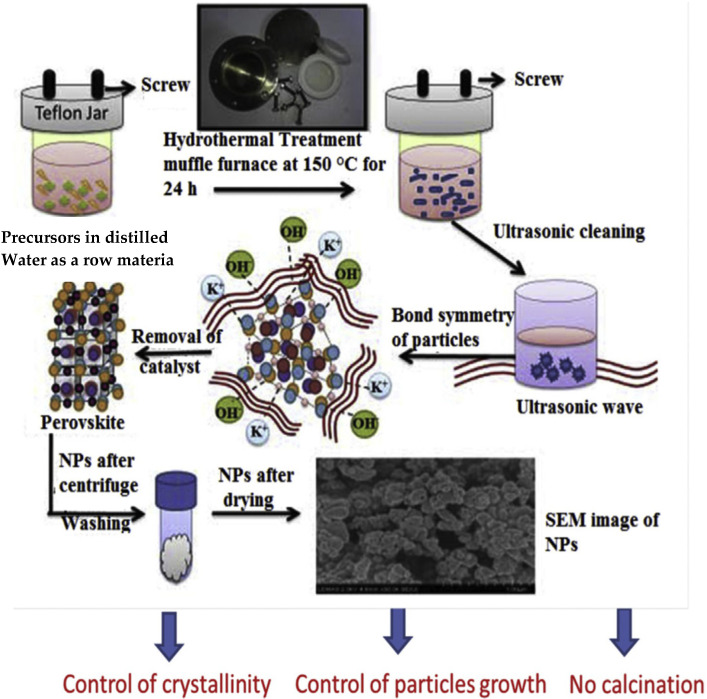
Scheme for the synthesis of perovskite using the hydrothermal process (Reprinted with permission from [101]).

**Figure 24 materials-15-03288-f024:**
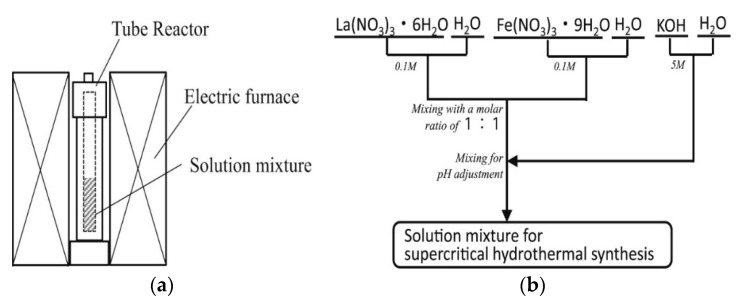
(**a**) Schematic diagram of a batch-type reactor vessel and (**b**) preparation of the solution mixture for supercritical hydrothermal synthesis (Reprinted with permission from [10]).

**Figure 25 materials-15-03288-f025:**
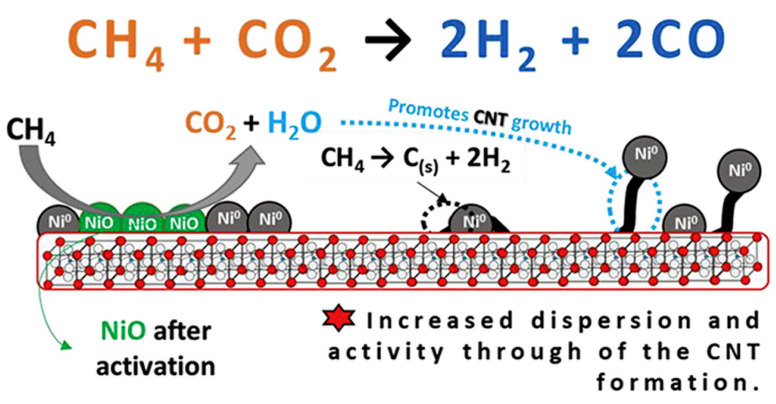
Schematic representation of the growth of carbon nanotubes on Ni/LaAlO_3_ with remaining NiO (Reprinted with permission from [19]).

**Figure 26 materials-15-03288-f026:**
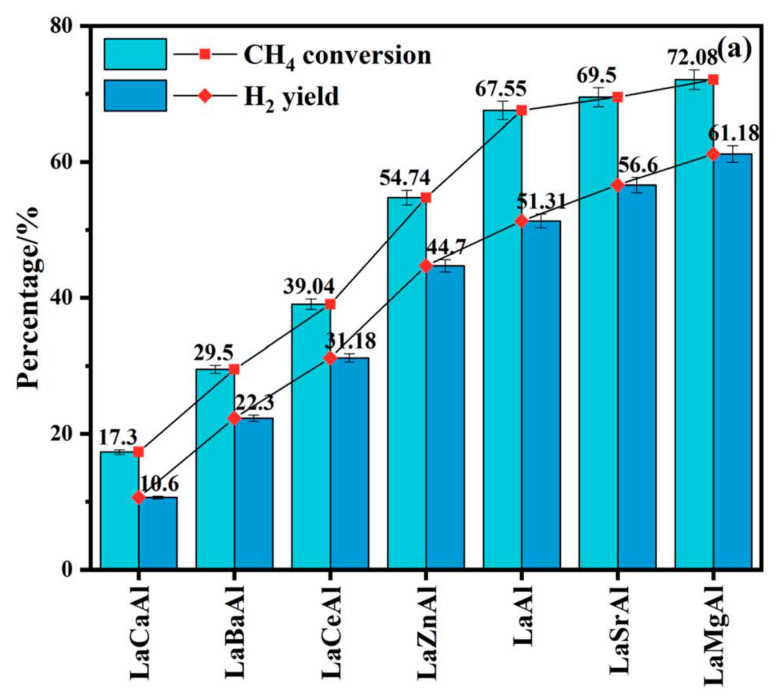
CH_4_ conversion and H_2_ yield for several A-site-substituted LaAlO_3_-supported Ni catalysts (Reprinted with permission from [104]).

**Figure 27 materials-15-03288-f027:**
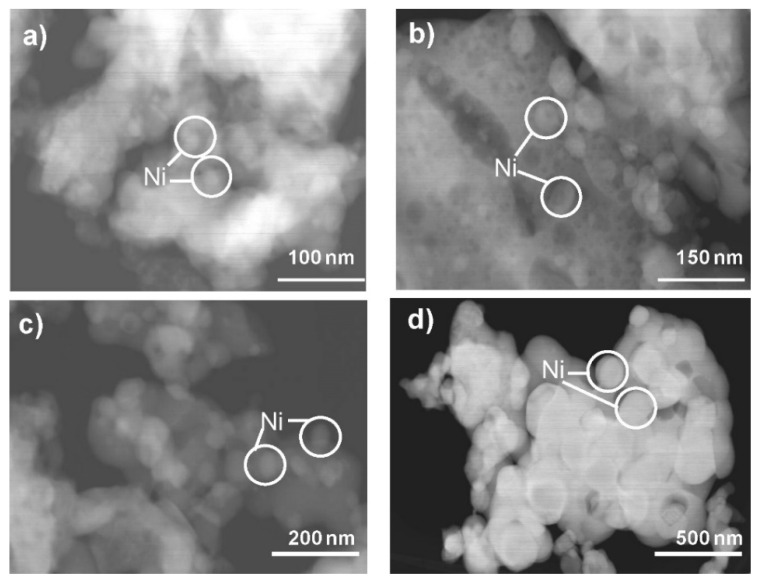
HAADF images for Ni/La_0.7_Sr_0.3_AlO_3−δ_ sintered at (**a**) 800 °C, (**b**) 900 °C, (**c**) 1000 °C, and (**d**) 1100 °C (Reprinted with permission from [112]).

**Figure 28 materials-15-03288-f028:**
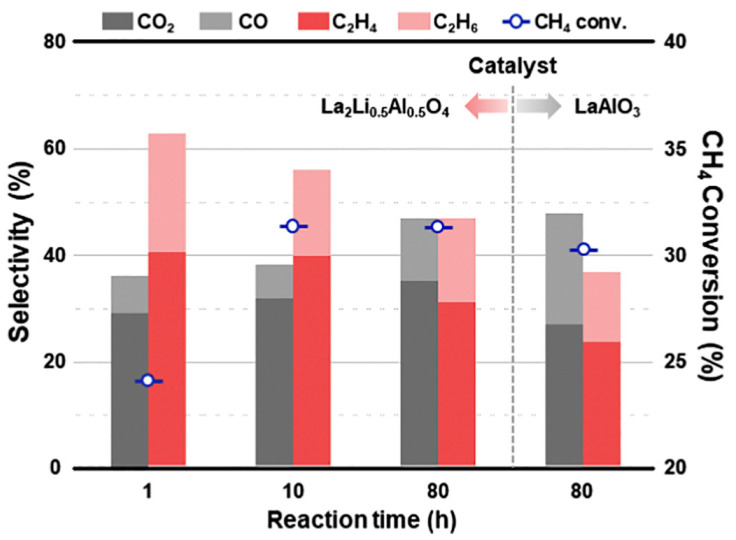
Catalytic activities of the La_2_Li_0.5_Al_0.5_O_4_ and LaAlO_3_ OCM catalysts (Reprinted with permission from [123]).

**Figure 29 materials-15-03288-f029:**
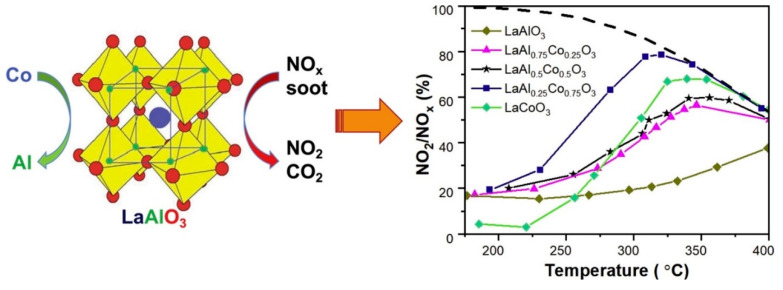
Structure of Co-doped LaAlO_3_ perovskite and the NO_2_ percentage for LaAl_1−x_Co_x_O_3_ calcined at 700 °C (Reprinted with permission from [20]).

**Table 1 materials-15-03288-t001:** Cations for ABO_3_-type perovskites, with their respective radii [7,22].

Dodecahedral A-Site			Octahedral B-Site		
Ion	Radius(Å)^a^	Radius(Å)^b^	Ion	Radius(Å)^a^	Radius(Å)^b^
Na^+^	1.06	1.32(IX)	Li^+^	0.68	0.74
K^+^	1.45	1.60	Cu^2+^	0.72	0.73
Rb^+^	1.61	1.73	Mg^2+^	0.66	0.72
Ag^+^	1.40	1.30(VIII)	Zn^2+^	0.74	0.75
Ca^2+^	1.08	1.35	Ti^3+^	0.76	0.67
Sr^2+^	1.23	1.44	V^3+^	0.74	0.64
Ba^2+^	1.46	1.60	Cr^3+^	0.70	0.62
Pb^2+^	1.29	1.49	Mn^3+^	0.66	0.65
La^3+^	1.22	1.32	Fe^3+^	0.64	0.64
Pr^3+^	1.10	1.14(VIII)	Co^3+^(LS)	-	0.52
Nd^2+^	1.09	1.12(VIII)	Co^3+^(HS)	0.63	0.61
Bi^3+^	1.07	1.11(VIII)	Ni^3+^(LS)	-	0.56
Ce^4+^	1.02	0.97(VIII)	Ni^3+^(HS)	0.62	0.60
Th^4+^	1.09	1.04(VIII)	Rh^3+^	0.68	0.66
			Ti^4+^	0.68	0.60
			Mn^4+^	0.56	0.54
			Ru^4+^	0.67	0.62
			Pt^4+^	0.65	0.63
			Nb^5+^	0.69	0.64
			Ta^5+^	0.69	0.64
			Mo^6+^	0.62	0.60
			W^6+^	0.62	0.60

**Table 2 materials-15-03288-t002:** Characteristics of methods for perovskites synthesis (adapted from [22,42,55,57,58,59]).

Synthesis Method	Surface Area (m^2^/g)	Particle Size and Extent of Agglomeration	Purity	Temperature of Crystallization (°C)	Advantages	Limitations
**Solid-state reaction**	<2.5	>1000 nm with moderate agglomeration	Very low	1100–1400	Cost effective, conventional, simplest, and operational simplicity.	Gives broad particle distribution as well as secondary phase formation.
**Co-precipitation**	5.5–20	>10 nm with high agglomeration	High	800	Control of size and shape of perovskites, simple and environmental friendly.	Lacks overall optimization, which could be attributed to the required controls during the washing step. Deficiency of metal cations.
**Sol–gel and Pechini process**	5–20	>10 nm with moderate agglomeration	Excellent	800–1000	High homogeneity and purity Accurate control of the composition of the final product	High temperature and long periods of time.
**Combustion**		>10 nm with low agglomeration	High	600–800	Highly pure, homogeneity and crystallinity	Production of large amount carbon in end product.
**Microwave assisted method**	1–36	>100 nm with low agglomeration	Excellent	600–800	Highly pure and avoiding particle coarsening. Time and energy saving.	Hard for scale-up and expensive equipment
**Hydrothermal and solvothermal routes**	~50	>100 nm with low agglomeration	Very high	No calcination	Can easily control morphology particle size, and crystallinity	Require high pressures (up to 15 MPa) inside autoclave

## Data Availability

Not applicable.

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
