# Peer review of "Progress and Recent Strategies in the Synthesis and Catalytic Applications of Perovskites Based on Lanthanum and Aluminum"

_materials, 2022, doi:10.3390/ma15093288_

Round 1

Reviewer 1 Report

In this work, Gil et. al., reviewed synthesis methods and catalytic applications of perovskites based La and Al. This manuscript is a comprehensive review of various synthesis techniques of LaAlO3 based perovskites and their characterization. The catalytic applications section is also well described.  However, there are some comments should be addressed before acceptance.

  1. Line 68, it should be cited more than one relevance reference, not only reference 8.
  2. Figure 1, there is typo “Hidrotermal and solvotermal”.
  3. Table 1, row Na+ and K+, at third column what are “1.32?” and “1.60?” ?
  4. The font in Figure 7, Figure 14, Figure 17 are not clear. For better vision, it would be better to rewrite all those tables.
  5. Line 367-369, how LaCO3OH and metal fluorides can reduce calcination temperature? The authors should provide more detail.
  6. Table 2, column limitation of co-precipitation method, what is the meaning of lacks overall optimization?
  7. Table 2, column advantages and limitations, there are some text in purple color such as “homogeneity and purity” , “product” and so on…
  8. Table 3 also presented some text in purple color.
  9. Figure 15, Figure 16, the left side of figures are cut-off, some of text and picture are missed.
  10. Line 709, please correct “Generaly”.
  11. Line 896, the authors wrote “(citric acid/metal = 2mol/mol). Is this 2:1 ratio?
  12. Line 1085, what are kinds of interesting optical properties of LaAlO3:Eu3+?
  13. Line 1298, it is better to use chemical name instead of C6H8O7.
  14. Line 1445, what is 7.8% y 11.5% more stable…?
  15. There are some typos in the manuscript, please check it carefully.

Author Response

Answer to Reviewer #1

First of all, we want to thank you for your detailed revision that helped us to improve the quality of our manuscript.Below please find the answers to your comments. The changes have been incorporated in the manuscript. In this work, Gil et. al., reviewed synthesis methods and catalytic applications of perovskites based La and Al. This manuscript is a comprehensive review of various synthesis techniques of LaAlO3 based perovskites and their characterization. The catalytic applications section is also well described. However, there are some comments should be addressed before acceptance. 

Q1) Line 68, it should be cited more than one relevance reference, not only reference 8. Reply: Other references than 8 have been considered (9,10,12,13 and 18).

Q2) Figure 1, there is typo “Hidrotermal and solvotermal”. Reply: The mistakes have been corrected.

Q3) Table 1, row Na+ and K+, at third column what are “1.32?” and “1.60?” ? Reply: Table 1 has been rewritten to clarify the information included.

Q4) The font in Figure 7, Figure 14, Figure 17 are not clear. For better vision, it would be better to rewrite all those tables. Reply: New figures have been included in the revised version.

Q5) Line 367-369, how LaCO3OH and metal fluorides can reduce calcination temperature? The authors should provide more detail. Reply: It has been reported that the calcination temperature to obtain LaAlO3 perovskite can be reduced using orthorhombic (o-)LaCO3OH instead of La2O3 powders and can also be reduced by adding metal fluorides. Fluoride ions play an important role in reducing the temperature formation of LaAlO3. However, the reason why this ocuurs is under investigation. Although, it has been reported the formation of transitory species (LaOF and LaF3), which could be favorably affecting the process.

Q6) Table 2, column limitation of co-precipitation method, what is the meaning of lacks overall optimization? Reply: Lacks overall optimization, which could be attributed to the required controls during the washing, as reported in reference 42.

Q7) Table 2, column advantages and limitations, there are some text in purple color such as “homogeneity and purity”, “product” and so on… Reply: The color in text has been revised.

Q8) Table 3 also presented some text in purple color. Reply: The color in text has been revised.

Q9) Figure 15, Figure 16, the left side of figures are cut-off, some of text and picture are missed. Reply: New figures have been included.

Q10) Line 709, please correct “Generaly”. Reply: The mistake has been corrected.

Q11) Line 896, the authors wrote “(citric acid/metal = 2mol/mol). Is this 2:1 ratio? Reply: Yes, the sentence has been rewritten.

Q12) Line 1085, what are kinds of interesting optical properties of LaAlO3:Eu3+? Reply: These materials showed interesting optical properties, such as emission very strong red luminescence and very efficient luminescence under the excitation of ultraviolet radiation. This information has been included in the revised version of the manuscript.

Q13) Line 1298, it is better to use chemical name instead of C6H8O7. Reply: The chemical name “citric acid” was used instead of C6H8O7.

Q14) Line 1445, what is 7.8% y 11.5% more stable…? Reply: Ni/LaAlO3 catalyst was 7.8 % and 11.5 % more stable than Ni catalyst supported on commercial α-Al2O3 for the conversion of methane and carbon dioxide, respectively. This sentence has been included in the revised version of the manuscript.

Q15) There are some typos in the manuscript, please check it carefully. Reply: The manuscript was carefully checked. 

Reviewer 2 Report

The authors have written a well-structured and thought review article. The background of the introduction is sufficiently described. The review listed a substantial up to date studies which are sufficient.

Only some minor revisions are still needed before accepting.

Page 2, line 51: structure.[8] Furthermore … should be: structure [8]. Furthermore …

Page 3, line 116: [23], → [23].

Page 7, Figure 7, in my opinion the descriptions on the figure are unclear .  Is it possible to correct them? please express more details.

Page 8, line 290: tenpearture → temperature,

Page how LaCO3OH and metal fluorides can reduce calcination temperature? The authors should provide more detail.

Page 13-14 and 16-17, (tables 2 and 3). Why some words in the tables are coloured (purple colour)?

Page 20, line 651, tempertures rangingf → temperatures ranging,

Page 21, line 702, shown th schematic → shown the schematic,

Page 25, Figure 17, also in this case the descriptions on the figure are unclear (hardly visible). Is it possible to correct them?

Page 25, line 895, LaAlO3:Bi3+(1 wt.%), there is no space between,

Page 46, line 1740: perovkite → perovskite.

Page 55  The discussion is superficial. The authors must give a proper discussion of the data by comparing to previous literature on the topic.

Author Response

Answer to Reviewer #2

First of all, we want to thank you for your detailed revision that helped us to improve the quality of our manuscript.Below please find the answers to your comments. The changes have been incorporated in the manuscript.

The authors have written a well-structured and thought review article. The background of the introduction is sufficiently described. The review listed a substantial up to date studies which are sufficient. Only some minor revisions are still needed before accepting.

Q1) Page 2, line 51: structure.[8] Furthermore … should be: structure [8].  Furthermore … Reply: The mistake has been corrected.

Q2) Page 3, line 116: [23], → [23]. Reply: The mistake has been corrected.

Q3) Page 7, Figure 7, in my opinion the descriptions on the figure are unclear (hardly visible).  Is it possible to correct them? Reply: Figure 7 has been revised in order to clarify the information included.

Q4) Page 8, line 290: tenpearture → temperature, Reply: The mistake has been corrected.

Q5) Page 13-14 and 16-17, (tables 2 and 3). Why some words in the tables are coloured (purple colour)? Reply: The color in text has been revised.

Q6) Page 20, line 651, tempertures rangingf → temperatures ranging, Reply: The mistake has been corrected.

Q7) Page 21, line 702, shown th schematic → shown the schematic, Reply: The mistake has been corrected.

Q8) Page 25, Figure 17, also in this case the descriptions on the figure are  unclear (hardly visible). Is it possible to correct them? Reply: Figure 17 has been revised in order to clarify the information included.

Q9) Page 25, line 855, LaAlO3:Bi3+(1 wt.%), there is no space between, Reply: The mistake has been corrected.

Q10) Page 46, line 1740: perovkite → perovskite. Reply: The mistake has been corrected.